# Human coronaviruses disassemble processing bodies

Mariel Kleer[1,2,3©], Rory P. Mulloy[1,2,3©], Carolyn-Ann Robinson[1,2,3], Danyel Evseev[1,2,3], Maxwell P. Bui-Marinos[1,2,3], Elizabeth L. Castle[4], Arinjay Banerjee[5,6,7,8], Samira Mubareka[8,9], Karen Mossman[10,11], Jennifer A. Corcoran[1,2,3]*

**1** Microbiology, Immunology and Infectious Diseases Department, University of Calgary, Calgary, Alberta, Canada, **2** Charbonneau Cancer Research Institute, University of Calgary, Calgary, Alberta, Canada, **3** Snyder Institute for Chronic Diseases, University of Calgary, Calgary, Alberta, Canada, **4** School of Biomedical Engineering, University of British Columbia, Vancouver, British Columbia, Canada, **5** Vaccine and Infectious Disease Organization, University of Saskatchewan; Saskatoon, Saskatchewan, Canada, **6** Department of Veterinary Microbiology, Western College of Veterinary Medicine, University of Saskatchewan; Saskatoon, Saskatchewan, Canada, **7** Department of Biology, University of Waterloo; Waterloo, Ontario, Canada, **8** Department of Laboratory Medicine and Pathobiology, University of Toronto, Toronto, Ontario, Canada, **9** Sunnybrook Research Institute, Toronto, Ontario, Canada, **10** Department of Medicine, Master University, Hamilton, Ontario, Canada, **11** Institute for Infectious Disease Research, McMaster University, Hamilton, Ontario, Canada

© These authors contributed equally to this work.
\* jennifer.corcoran@ucalgary.ca

**Data Availability Statement:** All relevant data are within the manuscript and its supporting files.

**Funding:** MK was supported by a Cumming School of Medicine graduate training award, a Canadian Institutes for Health Research (CIHR) CGS-M

## Abstract

A dysregulated proinflammatory cytokine response is characteristic of severe coronavirus infections caused by SARS-CoV-2, yet our understanding of the underlying mechanism responsible for this imbalanced immune response remains incomplete. Processing bodies (PBs) are cytoplasmic membraneless ribonucleoprotein granules that control innate immune responses by mediating the constitutive decay or suppression of mRNA transcripts, including many that encode proinflammatory cytokines. PB formation promotes turnover or suppression of cytokine RNAs, whereas PB disassembly corresponds with the increased stability and/or translation of these cytokine RNAs. Many viruses cause PB disassembly, an event that can be viewed as a switch that rapidly relieves cytokine RNA repression and permits the infected cell to respond to viral infection. Prior to this submission, no information was known about how human coronaviruses (CoVs) impacted PBs. Here, we show SARS-CoV-2 and the common cold CoVs, OC43 and 229E, induced PB loss. We screened a SARS-CoV-2 gene library and identified that expression of the viral nucleocapsid (N) protein from SARS-CoV-2 was sufficient to mediate PB disassembly. RNA fluorescent *in situ* hybridization revealed that transcripts encoding TNF and IL-6 localized to PBs in control cells. PB loss correlated with the increased cytoplasmic localization of these transcripts in SARS-CoV-2 N protein-expressing cells. Ectopic expression of the N proteins from five other human coronaviruses (OC43, MERS, 229E, NL63 and SARS-CoV) did not cause significant PB disassembly, suggesting that this feature is unique to SARS-CoV-2 N protein. These data suggest that SARS-CoV-2-mediated PB disassembly contributes to the dysregulation of proinflammatory cytokine production observed during severe SARS-CoV-2 infection.

scholarship, and a CIHR doctoral award and RPM was supported by a Snyder Institute Beverley Phillips Doctoral training award and a CIHR doctoral award. This study was supported by operating funds awarded to JAC from the Canadian Institutes for Health Research: a COVID rapid response operating grant (#177704) and an operating grant (#175622) to the Coronavirus Variants Rapid Response Network (CoVaRR-Net), of which JAC is a member. The funders had no role in study design, data collection and analysis, decision to publish, or preparation of the manuscript.

**Competing interests:** The authors have declared that no competing interests exist.

## Author summary

Processing bodies (PBs) are cytoplasmic RNA and protein granules that regulate gene expression post-transcriptionally because they degrade or suppress cellular mRNA transcripts. PBs are particularly relevant for the regulation of inflammatory cytokine production because many cytokine RNAs contain sequences that direct them to PBs for rapid turnover or suppression. When PBs disassemble, this increases capacity for cytokine translation. Here, we show PB loss is a conserved feature of CoV infection, including SARS-CoV-2, and that expression of the SARS-CoV-2 nucleocapsid (N) protein is sufficient to mediate PB disassembly. N protein expression also alters the distribution and abundance of inflammatory cytokine transcripts that are typically contained within PBs, thereby returning them to the cytoplasmic pool of mRNA available for translation. Together, these data underscore the importance of PB loss to CoV infection and suggest it may contribute to the aberrant inflammatory nature associated with severe CoV infections.

## Introduction

Processing bodies (PBs) are membraneless ribonucleoprotein (RNP) granules found in the cytoplasm of all cells [1,2]. PBs control cellular gene expression because they either degrade or sequester RNA transcripts, preventing their translation into protein. PBs contain many enzymes required for mRNA turnover, including those needed for decapping (Dcp2 and co-factors Dcp1a and Edc4/Hedls) and decay of the RNA body (5'-3' exonuclease Xrn1 and RNA helicase Rck/DDX6) as well as components of the RNA-induced silencing complex [3,4]. Not all coding RNAs are regulated by PBs, but those that are typically encode potent regulatory molecules like growth factors, pro-inflammatory cytokines, and angiogenic factors. One group of protein-coding mRNAs commonly found in PBs bear destabilizing AU-rich elements (AREs) in their 3'-untranslated regions (3'-UTRs) and include most proinflammatory cytokine transcripts [5–7]. These RNAs shuttle to PBs by virtue of interactions between the AU-rich element and RNA-binding proteins (RBPs) [3,8–11]. We and others have showed that the presence of PBs correlates with increased turnover/suppression of ARE-mRNAs [7,12–15]. Conversely, when PBs are lost, constitutive ARE-mRNA suppression is reversed, and ARE-mRNA transcripts and/or their translation products accumulate. Therefore, PB disassembly can be viewed as a switch that permits cells to rapidly respond and translate ARE-containing proinflammatory cytokine RNA into molecules such as IL-6, IL-8, IL-1β, and TNF [5]. PBs provide an extra layer of post-transcriptional control enabling the cell to fine-tune the production of potent molecules like proinflammatory cytokines.

Although PBs are constitutive, they are also dynamic, changing in size and number in response to different stimuli or stressors. This dynamic disassembly/assembly is possible because PBs behave as biomolecular condensates that form via liquid-liquid phase separation of proteins [16–19]. PBs form via sequential multivalent RNA-protein interactions, with a small group of proteins that contain regions of intrinsic disorder serving as the essential scaffold onto which additional proteins or RNA can be recruited as the PB matures [9,16,20–24]. Despite the recognition of PBs as dynamic entities, our understanding of the signals that induce PB disassembly remains incomplete. We and others have shown that stressors which activate the p38/MK2 MAP kinase pathway, as well as many virus infections elicit PB disassembly [12,13,15,25,26]. Disassembly can occur by a direct interaction between a viral protein (s) and a PB component that is subsequently re-localized to viral replication and transcription

compartments (vRTCs) [27–29] or cleaved by viral proteases [30–32]. Viruses can also cause PB disassembly indirectly by activating p38/MK2 signaling [12,13,26].

There are numerous reports of viral gene products that trigger PB disassembly, yet corresponding reports of viral gene products that stimulate PB formation are rare, suggesting that PBs possess direct antiviral function and their disassembly may favour viral replication in ways that we do not yet grasp [33,34]. Even though other RNPs, such as stress granules, have emerged as important components of our antiviral defenses that contribute to sensing virus and triggering innate immune responses, evidence to support a direct antiviral role for PBs is less well established [34–37]. Such a role has been defined for several PB-localized enzymes that impede viral replication (e.g. APOBEC3G, MOV10); however, in these cases, the mechanism of viral restriction was attributed to the enzymatic activity of the PB protein(s) whereas its localization to PBs was not deemed as significant [28,29,31,38–45]. Nonetheless, the disassembly of PBs by diverse viruses strongly suggests they negatively regulate virus replication. The reason viruses disassemble PBs may be to limit their antiviral activity; however, because PBs also control turnover/suppression of many proinflammatory cytokine transcripts, their disruption by viruses contributes to high proinflammatory cytokine levels, alerting immune cells to the infection.

The family *Coronaviridae* includes seven viruses that infect humans, including the four circulating 'common cold' coronaviruses (CoVs), CoV-OC43, CoV-229E, CoV-NL63, and CoV-HKU1 and three highly pathogenic viruses that cause severe disease in humans: MERS-CoV, SARS-CoV, and the recently emerged SARS-CoV-2 [46–51]. Severe COVID-19 is characterized by aberrant proinflammatory cytokine production, endothelial cell (EC) dysfunction and multiple organ involvement [52–64]. Even with intense study, we do not yet appreciate precisely how SARS-CoV-2 infection causes severe COVID-19 in some patients and mild disease in others, though a mismanaged or delayed IFN response and an overactive cytokine response is thought to underlie severe outcomes [65–70]. Despite some contrasting reports [71–73], what is clear is that SARS-CoV-2 proteins use a multitude of mechanisms to outcompete cellular antiviral responses [68,74–85].

To determine if SARS-CoV-2 and other CoVs interact with PBs to alter the cellular antiviral response, we performed an analysis of PBs after CoV infection. Prior to this submission, no published literature was available on human CoVs and PBs, and only two previous reports mentioned PB dynamics after CoV infection. Murine hepatitis virus (MHV) was reported to increase PBs at early infection times, while transmissible gastroenteritis coronavirus (TGEV) infected cells displayed complete PB loss by 16 hours post infection [86,87]. Observations that SARS-CoV-2 infection induced elevated levels of many PB-regulated cytokines, such as IL-6, IL-10, IL-1β and TNF [53,54,57,69,70] suggested that human CoVs like SARS-CoV-2 may reshape the cellular innate immune response in part by targeting PBs for disassembly. We now present the first evidence to show that three human CoVs, including SARS-CoV-2, trigger PB disassembly during infection. By screening a SARS-CoV-2 gene library, we identified that the nucleocapsid (N) protein was sufficient for PB disassembly. However, this feature is not common to all human coronavirus N proteins, as overexpression of MERS-CoV-N, OC43-N, 229E-N and NL63-N was insufficient to induce PB loss and expression of SARS-CoV-1 N protein displayed an intermediate phenotype. SARS-CoV-2 N protein-mediated PB disassembly also correlated with the redistribution of two PB-regulated cytokine transcripts, TNF and IL-6, from PBs to the cytoplasm and an increase in abundance of TNF RNA. These data suggest that PB disassembly by SARS-CoV-2 N may promote the redistribution of certain cytokine transcripts from PB foci to the cytoplasm. Taken together, these results show that PBs are targeted for disassembly by human CoV infection, and that this phenotype may contribute in part to reshaping cytokine responses to SARS-CoV-2 infection.

## Results

### Infection with human coronaviruses causes PB loss

Endothelial cells (ECs) have emerged as playing a significant role in severe COVID; as sentinel immune cells they are important sources for many of the cytokines elevated in severe disease and are infected by SARS-CoV-2 *in vivo* [56,58–60,88]. However, others have shown that commercial primary human umbilical vein endothelial cells (HUVECs) require ectopic expression of the viral receptor, ACE2, to be susceptible to SARS-CoV-2 [89]. We recapitulated those findings and showed that after HUVECs were transduced with an ACE2-expressing lentivirus (HUVEC[ACE2]), they were permissive to SARS-CoV-2 (Wuhan-like ancestral Toronto isolate; TO-1) [90] (S1A Fig). To use HUVEC[ACE2] for studies on PB dynamics, we confirmed that ACE2 ectopic expression had no effect on PB number in HUVECs (S1B Fig). Confirming this, we infected HUVEC[ACE2] with SARS-CoV-2 (MOI = 3) to determine if PBs were altered. SARS-CoV-2 infected cells were identified by immunostaining for the viral nucleocapsid (N) while PBs were identified by immunostaining for two different PB resident proteins, the RNA helicase DDX6, and the decapping cofactor, Hedls. PBs, measured by staining for both markers, were absent in most SARS-CoV-2 infected HUVECs[ACE2] by 24 hours post infection (Fig 1A–1D). We quantified the loss of cytoplasmic puncta using a method described previously [91] and showed that by 24 hours post infection, SARS-CoV-2 infected cells displayed a significant reduction in PBs compared to mock-infected controls (Fig 1B and 1D).

To confirm that PBs were reduced by SARS-CoV-2 infection of naturally permissive cells derived from respiratory epithelium, we infected Calu-3 cells with SARS-CoV-2. Infected cells were identified by immunostaining for N protein 24 hours post infection and PBs were stained for DDX6 and Hedls. We observed PB loss in most but not all infected cells (Fig 1E–1H). We quantified PB loss and showed that by 24 hours post infection, SARS-CoV-2 infected Calu-3 cells also displayed a significant reduction in PBs compared to mock-infected controls (Fig 1F and 1H). As the SARS-CoV-2 pandemic has progressed, new variants of concern (VOCs) have continued to emerge [92,93]. To determine if VOCs also induced PB disassembly, we infected HUVEC[ACE2] with SARS-CoV-2 VOCs Alpha, Beta, Gamma and Delta (MOI = 2) and compared PB disassembly to an ancestral isolate of SARS-CoV-2 (Wuhan-like Toronto isolate; TO-1) (Fig 2). Infected cells were identified by immunostaining for N protein 24 hours post infection and PBs were identified using Hedls. Significant PB disassembly was observed for all VOCs, although we noted slightly less PB disassembly mediated by TO-1 compared to the experiments in Fig 1, likely due to the decrease in MOI between these two experiments (Fig 2).

To determine if PBs were lost in response to infection with other human coronaviruses, we established infection models for the *Betacoronavirus*, OC43, and the *Alphacoronavirus*, 229E. We found HUVECs were permissive to both OC43 and 229E (S2A–S2B Fig). We then performed a time-course experiment wherein OC43-infected HUVECs were fixed at various times post infection and immunostained for the viral N protein and the PB-resident protein DDX6. We observed that PBs were largely absent in OC43 N protein-positive cells but present in mock-infected control cells (Fig 3A and 3B). 229E-infected HUVECs were stained for DDX6 to measure PBs and for dsRNA to denote infected cells due to a lack of commercially available antibodies for 229E. CoV infected cells are known to form an abundance of dsRNA due to viral replication and transcription from a positive-sense RNA genome making this a suitable marker for virally infected cells [94]. In parallel, we confirmed 229E infection by performing RT-qPCR for viral genomic and subgenomic RNA (S2C–S2D Fig). After 229E infection, we also found that PBs were significantly reduced (Fig 3C and 3D). Because of antibody incompatibility, we were unable to co-stain infected cells for the PB protein Hedls and OC43 N protein or dsRNA. In lieu of this, we performed additional OC43 and 229E infections and

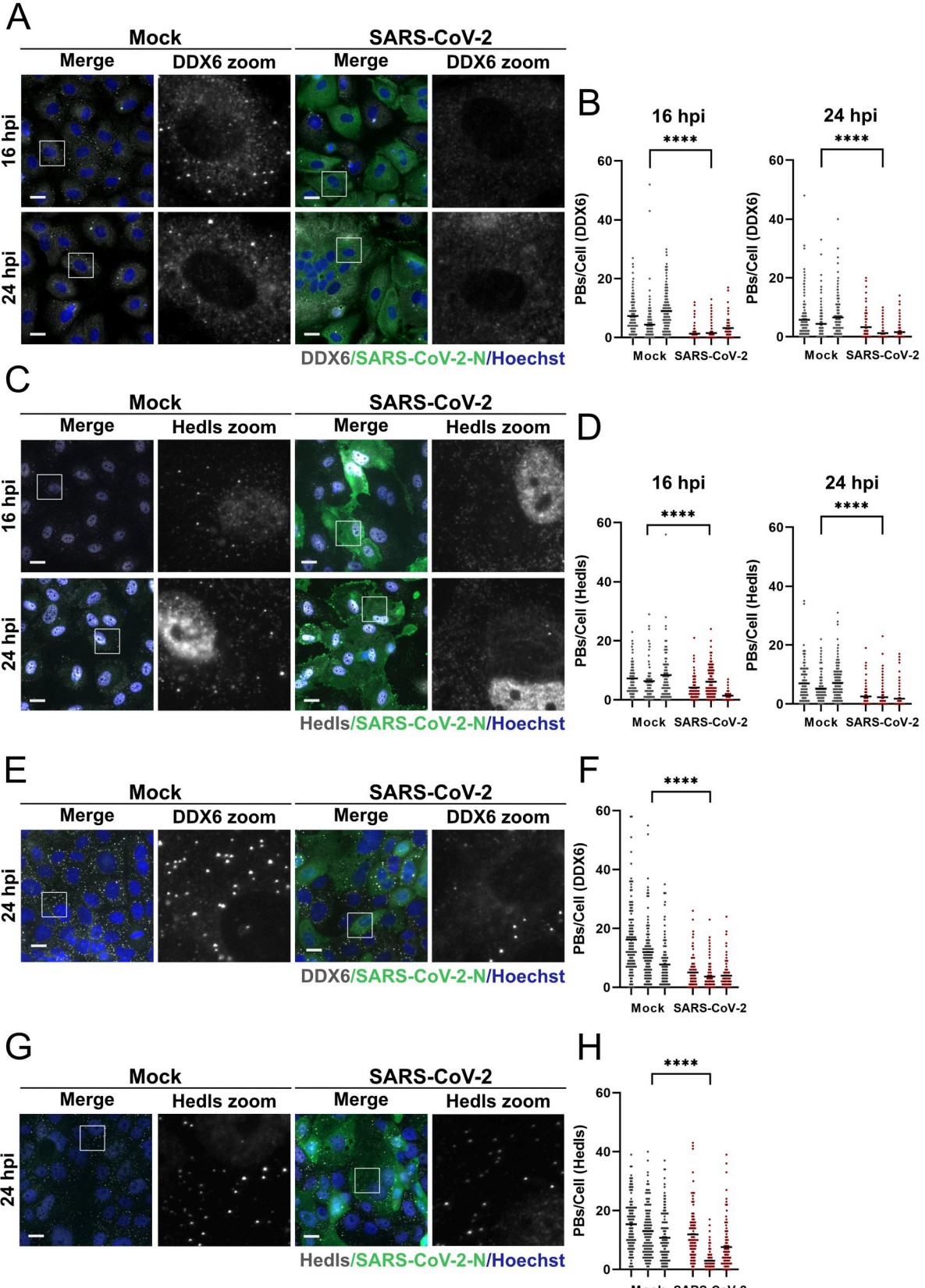

**Fig 1. Processing bodies are absent in SARS-CoV-2 infected cells. A-D.** HUVECs were transduced with recombinant lentiviruses expressing human ACE2 (HUVEC$^{ACE2}$), selected, and infected with SARS-CoV-2 TO-1 isolate at an MOI of 3 or a time-matched mock infection control. At 16 or 24 hours post infection, cells were fixed and immunostained for SARS-CoV-2 N protein (green; Alexa488) and DDX6 (A, white; Alexa555) or Hedls (C, white; Alexa555). Nuclei were stained with Hoechst (blue). Representative images from one of three independent experiments are shown (A, C). PBs were quantified using CellProfiler by measuring DDX6 puncta (B) and Hedls puncta (D) in SARS-CoV-2-infected cells (thresholded by N protein staining) or in mock-infected cells. These data represent three independent biological replicates ($n = 3$) with >80 cells measured per condition (mock and infected) per replicate. Each mock and infected replicate pair plotted independently (B, D); mean (****, $p < 0.0001$). **E-H.** Calu3 cells were infected with SARS-CoV-2 TO-1 isolate (MOI = 3) or time-matched mock-infected control. 24 hours post infection cells were fixed and immunostained for SARS-CoV-2 N (green; Alexa488), DDX6 (E; white; Alexa555) or Hedls (G; white; Alexa555). Nuclei were stained with Hoechst (blue). PBs were quantified using CellProfiler as in B and D (F, H). These data represent three independent biological replicates ($n = 3$) with >130 cells measured per condition (mock and infected) per replicate. Representative images from one experiment of three are shown (E, G) and each mock and infected replicate pair plotted independently (F, H); mean (****, $p < 0.0001$). Statistics were performed using a Mann-Whitney rank-sum test. Scale bar = 20 µm.

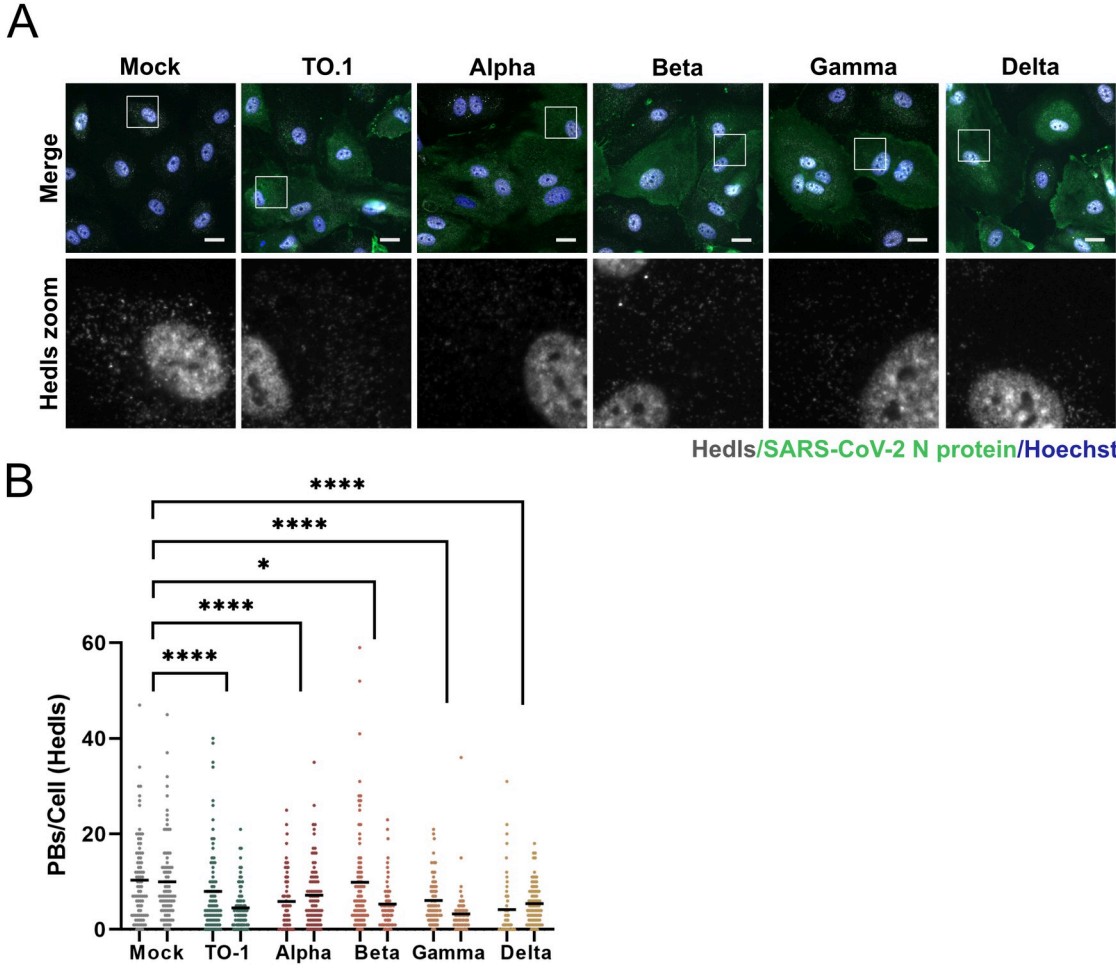

**Fig 2. Processing bodies are absent in SARS-CoV-2 Alpha, Beta, Gamma, and Delta variant infected cells. A.** HUVECs were transduced with recombinant lentiviruses expressing human ACE2 (HUVEC$^{ACE2}$), selected and infected with SARS-CoV-2 TO-1, Alpha, Beta, Gamma, or Delta isolates (MOI = 2), or a mock infection control. 24 hours post infection cells were fixed and immunostained for SARS-CoV-2 N protein (green; Alexa488) and Hedls (white; Alexa555). Nuclei were stained with Hoechst (blue). Representative images from one of two independent experiments are shown. **B.** PBs were quantified as in Fig 1. These data represent two independent biological replicates ($n = 2$) with >80 cells measured per condition (mock and infected) per replicate. Each mock and infected replicate pair plotted independently; mean. Statistics were performed using Kruskal-Wallis *H* test with Dunn's correction (*, $p < 0.0332$; ****, $p < 0.0001$). Scale bar = 20 µM.

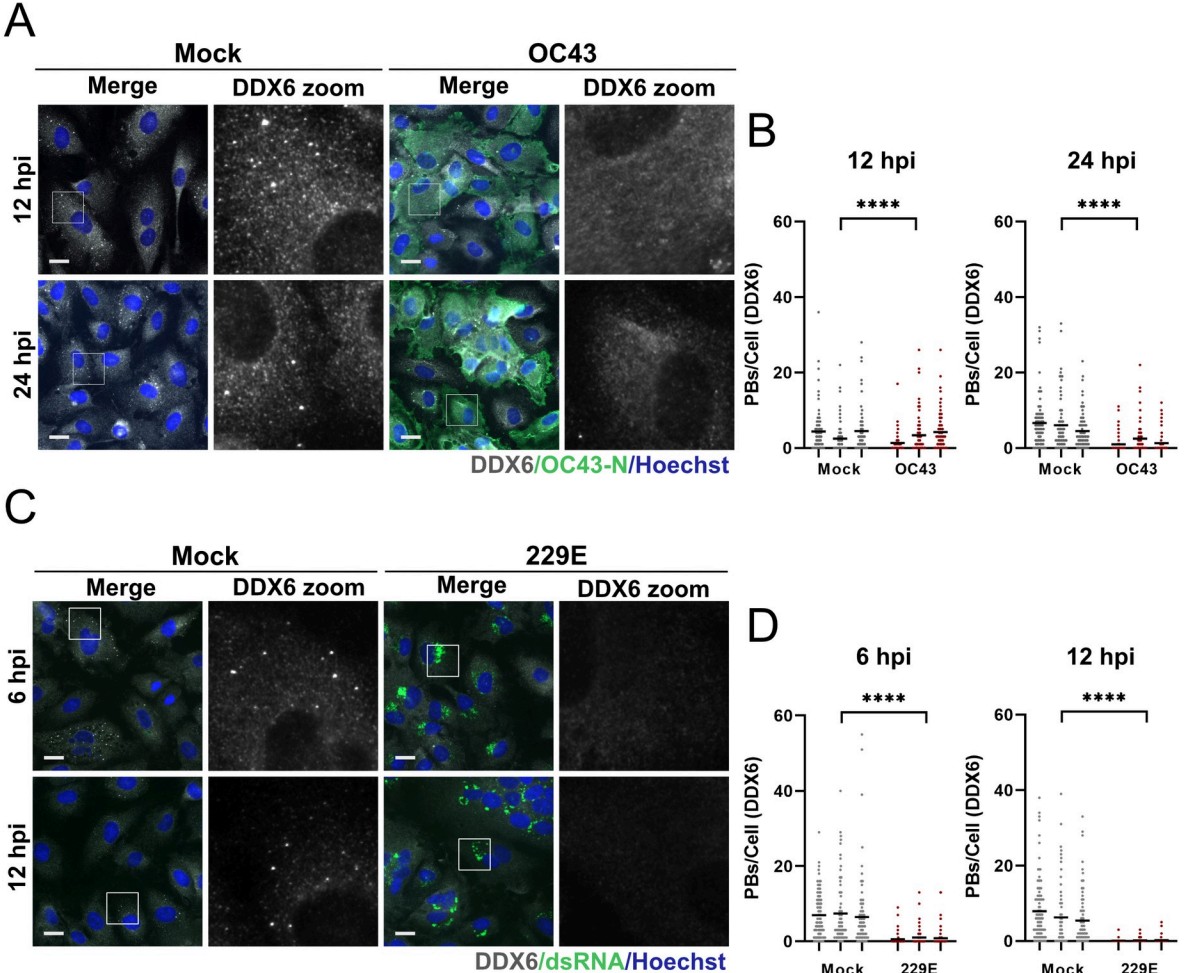

**Fig 3. Processing bodies are absent in OC43 and 229E infected cells. A-B.** HUVECs were infected with OC43 (TCID$_{50}$ = 2 x 10$^4$) or mock-infected. Cells were fixed at 12 or 24 hours post infection and immunostained for DDX6 (PBs; white; Alexa555) and OC43 N protein (green; Alexa488). Nuclei were stained with Hoechst (blue). These data represent three independent biological replicates (*n = 3*) with >80 cells measured per condition (mock and infected) per replicate. Representative images from one of three experiments are shown. DDX6 puncta in mock or OC43-infected cells were quantified as in Fig 1. Each mock and infected replicate pair plotted independently; mean. **C-D.** HUVECs were infected with 229E (TCID$_{50}$ = 2.4 x 10$^3$) or mock-infected. Cells were fixed at 6 or 12 hours post infection and immunostained for DDX6 (PBs; white; Alexa555) and dsRNA (proxy for 229E infection; green; Alexa488). Nuclei were stained with Hoechst (blue). These data represent three independent biological replicates (*n = 3*) with >80 cells measured per condition (mock and infected) per replicate. Representative images from one of three experiments are shown. DDX6 puncta in mock or 229E-infected (dsRNA+) cells were quantified as in Fig 1. Each mock and infected replicate pair plotted independently; mean. Statistics were performed using a Mann-Whitney rank-sum test (****, $p < 0.0001$). Scale bar = 20 μM.

co-stained infected cells using antibodies for Hedls and DDX6 (S3 Fig). We then performed additional quantification of PB loss using the Hedls marker to label PBs. These data also show robust PB loss in response to infection with OC43 and 229E (S3 Fig).

PBs will disassemble if key scaffolding proteins are lost; these include the RNA helicase DDX6, the translation suppressor 4E-T, the decapping cofactors Hedls/EDC4 and DCP1A, and the scaffolding molecule Lsm14A [95]. To elucidate if CoV-infected cells displayed decreased steady-state levels of PB resident proteins, we immunoblotted infected cell lysates for PB proteins XRN1, DCP1A, or DDX6, and Hedls (Fig 4) and quantified protein levels by densitometry (S4 Fig). SARS-CoV-2 infection of HUVEC$^{ACE2}$ cells did not alter steady-state levels of these proteins compared to uninfected cells (Fig 4A). OC43-infected HUVECs

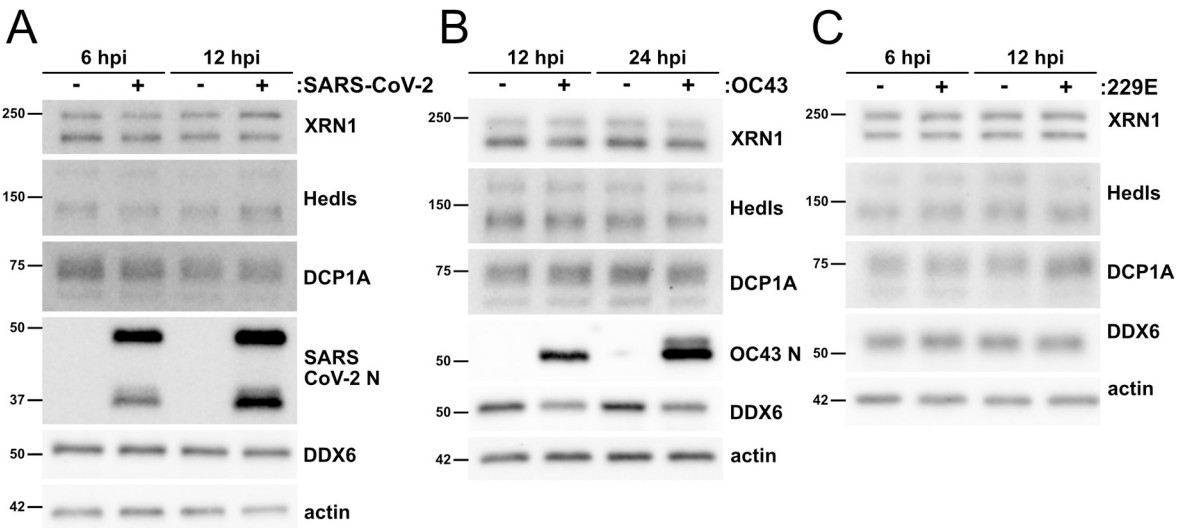

**Fig 4. Coronavirus infection does not alter steady state levels of most processing body proteins. A.** HUVECs were transduced with human ACE2 (HUVEC$^{ACE2}$), selected, and infected with SARS-CoV-2 TO-1 isolate (MOI = 3). Cells were lysed at 6 and 12 hours post infection and immunoblotting was performed using XRN1, Hedls, DCP1A, DDX6, SARS-CoV-2 N, and β-actin specific antibodies. One representative experiment of two is shown. **B-C.** HUVECs were infected with OC43 (B, TCID$_{50}$ = 2 x 10$^4$) or 229E (C, TCID$_{50}$ = 2.4 x 10$^3$). Cells were lysed at 12 and 24 hours post infection (B, OC43) or 6 and 12 hours post infection (C, 229E). Immunoblotting was performed using XRN1, Hedls, DCP1A, DDX6, OC43 N protein (B only), and β-actin specific antibodies. One representative experiment of three is shown.

displayed comparable levels of XRN1, DCP1A, and Hedls relative to uninfected cells; however, OC43 infection decreased steady-state levels of DDX6 (Fig 4B). 229E-infected HUVECs showed no significant change in PB protein expression after infection compared to controls (Fig 4C).

PBs are important sites for the post-transcriptional control of inflammatory cytokine transcripts containing AU-rich elements, and PB loss correlates with enhanced levels of some of these transcripts [7,12–15]. To determine if ARE-mRNAs are elevated, and therefore subject to this type of regulation during CoV infection, we harvested total RNA from SARS-CoV-2-, OC43- or 229E-infected HUVECs and performed RT-qPCR for five ARE-containing cytokine transcripts, IL-6, CXCL8, COX-2, GM-CSF, and IL-1β (Fig 5A–5C). We attempted to quantify TNF RNA as well, but the low copy number (high Ct value) of TNF RNA in mock control samples made quantification by RT-qPCR inaccurate and we were unable to measure the fold-increase of TNF RNA after CoV infection. We observed increased levels of three ARE-containing transcripts, IL-6, CXCL8, and COX-2, compared to uninfected cells, particularly in SARS-CoV-2-infected HUVEC$^{ACE2}$ cells (Fig 5A). Taken together, these data indicate that infection with human CoVs including SARS-CoV-2 induces PB loss, and that some PB-regulated cytokine ARE-mRNAs are elevated during CoV infection.

## A screen of SARS-CoV-2 genes reveals mediators of PB loss

The genome of SARS-CoV-2 is predicted to contain up to 14 open reading frames (ORFs). The two 5' proximal ORFs (1a and 1ab) encode two large polyproteins which are processed by viral proteases into 16 non-structural proteins (nsp1-16) essential for viral genome replication and transcription [46]. The 3' end of the SARS-CoV-2 genome is predicted to code for ORFs that are expressed from 9 subgenomic mRNAs [96]. Among these are the four structural proteins spike (S), envelope (E), membrane (M) and nucleocapsid (N) and up to 9 potential

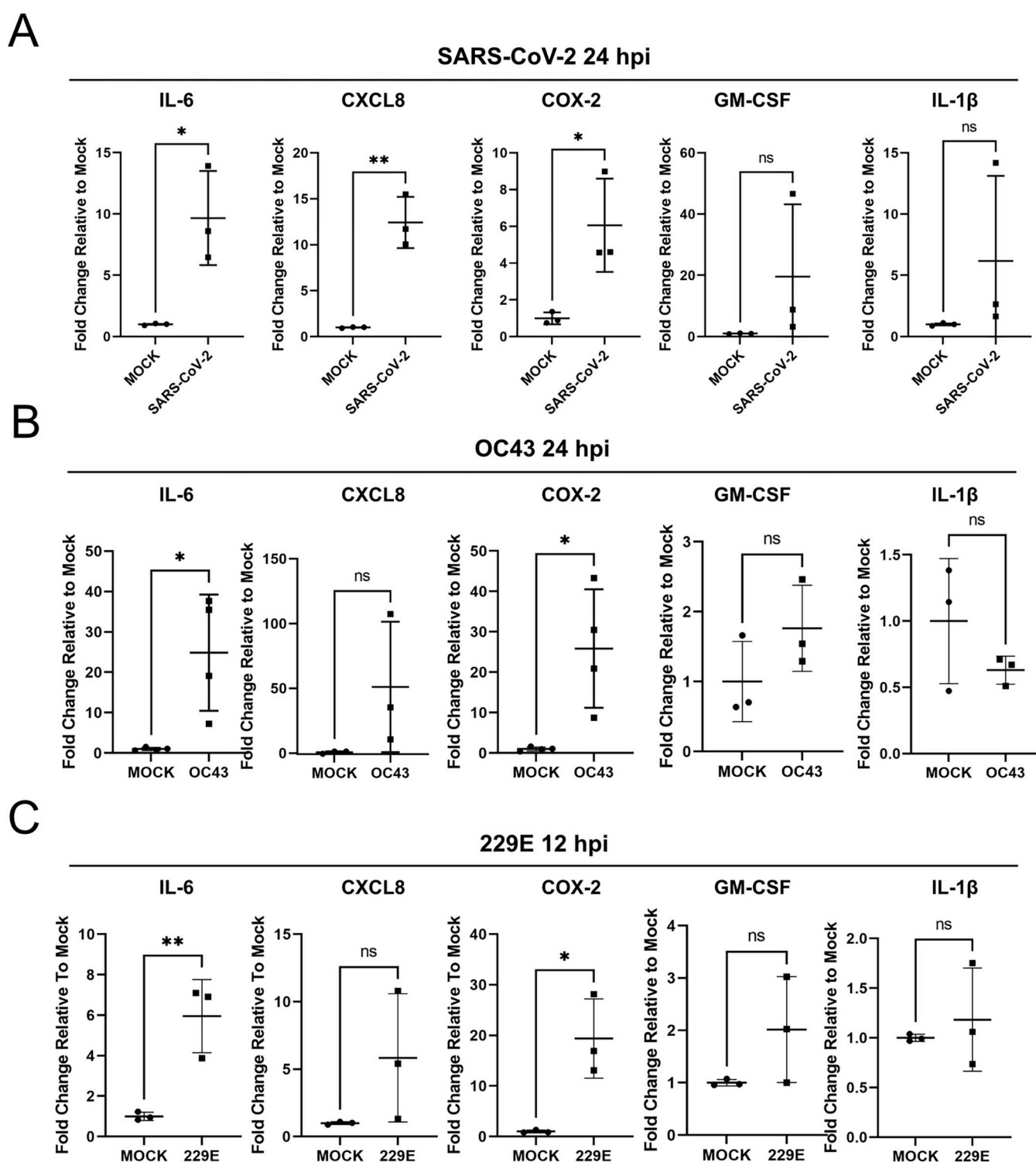

**Fig 5. Steady state levels of selected ARE-mRNAs are elevated during coronavirus infection. A.** HUVECs were transduced with recombinant lentiviruses expressing human ACE2 (HUVEC$^{ACE2}$), selected, and infected with SARS-CoV-2 TO-1 isolate (MOI = 3). RNA was harvested 24 hours post infection and RT-qPCR was performed using IL-6, CXCL8, COX-2, GM-CSF, IL-1β and HPRT (house-keeping gene) specific primers. Values are represented as fold change relative to mock-infection. $n = 3$; mean ± SD ($*$, $p < 0.05$; $**$, $p < 0.01$; ns, nonsignificant). **B-C.** HUVECs were infected with OC43 (B, TCID$_{50}$/mL = 3.5 x 10$^4$) or 229E (C, TCID$_{50}$/mL = 1 x 10$^{3.24}$). RNA was harvested 24 and 12 hours post infection for OC43 (B) and 229E (C), respectively, and RT-qPCR was performed as in (A). Values are represented as fold change relative to mock-infection. Statistics were performed using an unpaired Student's $t$-test. $n \geq 3$; mean ± SD ($*$, $p < 0.05$; $**$, $p < 0.01$; ns, nonsignificant).

accessory proteins, not all of which have been validated as expressed in infected cells [96]. To test which SARS-CoV-2 gene product(s) was responsible for PB disassembly, we obtained a plasmid library of 27 SARS-CoV-2 genes from the Krogan lab; this library included 14 nsps (excluding nsp3 and nsp16), all structural (S, E, M, N) and candidate accessory genes (ORFs 3a, 3b, 6, 7a, 7b, 8, 9b, 9c, 10) [96]. We individually transfected each plasmid and immunostained for each of the SARS-CoV-2 proteins using antibodies to the Strep-tag II and for PBs using anti-DDX6 (Figs 6A and S5). Relative to control cells, many SARS-CoV-2 ORF transfected cells still displayed DDX6-positive PBs; however, expression of some SARS-CoV-2 genes reduced DDX6-positive puncta, including N and ORF7b (Fig 6A) to a similar or greater extent than our positive control, the KapB protein from Kaposi's sarcoma-associated herpesvirus, which we have previously shown causes PB disassembly (Fig 6B and 6C) [13,91,97,98]. We quantified the number of DDX6-positive PBs per cell for each transfection as in [91] and found that the average number of PBs per cell was reduced relative to our negative control after transfection of eight SARS-CoV-2 genes: nsp7, ORF7b, N, ORF9b, ORF3b, nsp6, nsp1, and nsp11 (Fig 6D and 6E). This quantification was performed in two different ways. In most cases, transfected cells were identified by co-staining for the Strep-tag II fused to each gene (Figs 6A and S5). In such cases, we quantified DDX6-positive puncta only in those cells that were transfected and not in bystander cells (Fig 6D, thresholded). These data identified three SARS-CoV-2 proteins that may cause PB loss in a cell autonomous manner: nsp7, ORF7b and N (Fig 6D). For the remaining transfections (nsp1, nsp5, nsp6, nsp11, nsp13, nsp14, ORF3b, ORF6, ORF9b, ORF9c) immunostaining for the Strep-tag II was not robust and we were unable to threshold our PB counts (S5 Fig). In these samples, we quantified PBs in all cells of the monolayer (Fig 6E, unthresholded). These data identified five additional SARS-CoV-2 proteins from our screen that may cause PB loss: ORF9b, ORF3b, nsp1, nsp6 and nsp11 (Fig 6E). We verified the expression of all constructs, including low expressors (nsp1, nsp5, nsp6, nsp11, nsp13, nsp14, ORF3b, ORF6, ORF9b and ORF9c) by immunoblotting whole cell lysates harvested from parallel transfections (Fig 6F). We were unable to detect nsp4, ORF9c or ORF10 by immunoblotting; however, we did visualize expression of these proteins by immunostaining (Figs 6F and S5). We eliminated low confidence hits (nsp7, ORF9b) and low expressors (nsp6, nsp11) from further studies and proceeded with validation of the top four hits (ORF7b, N, ORF3b, nsp1).

## The nucleocapsid protein of SARS-CoV-2 induces PB disassembly

We tested four top hits from our PB screen in more relevant endothelial cells as these cells can be infected, express inflammatory cytokines, and stain robustly for PBs. HUVECs were transduced with recombinant lentiviruses expressing N, nsp1, ORF3b, and ORF7b or empty vector control lentiviruses. We also included recombinant lentiviruses expressing nsp14 in this experiment because of its exoribonuclease activity and ability to diminish cellular translation and interferon responses [84]. Transduced cells were selected for transgene expression and then fixed and stained for the endogenous PB marker protein DDX6 and for the Strep-tag II on each of SARS-CoV-2 constructs. We observed robust staining of the viral nucleocapsid (N) protein in the transduced cell population (Fig 7A) but were unable to detect expression of nsp1, nsp14, ORF3b or ORF7b by immunostaining (S6A Fig). We quantified PB loss in the selected cells and observed decreased PB numbers in cell populations expressing N, nsp14, ORF3b and ORF7b; however, the most robust PB loss was induced in N-expressing cells, which displayed a significant reduction in PB numbers as well as strong immunostaining (Figs 7B and S6). We were concerned that we could not detect the other four transgenes by immunostaining; therefore, we performed immunoblotting for the Strep-II tag on lysates from each

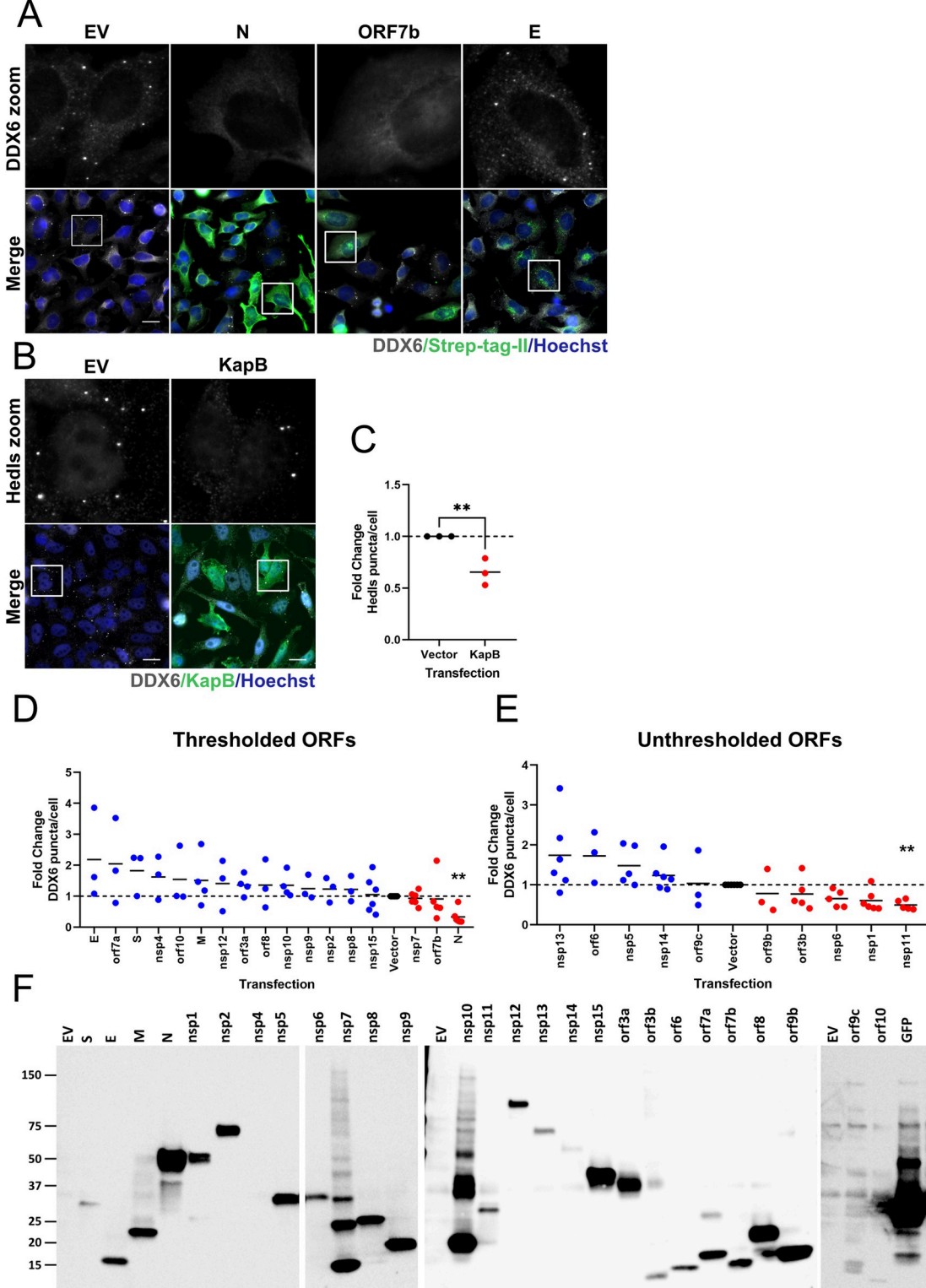

**Fig 6. Identification of SARS-CoV-2 ORFs that mediate processing body loss. A.** HeLa cells expressing GFP-Dcp1a were transfected with an empty vector (EV) or 2X Strep-tagged SARS-CoV-2 ORFs for 48 hours then fixed and immunostained for Strep-tag II (viral ORF; green; Alexa647) or DDX6 (PBs; white; Alexa555). Nuclei were stained with Hoechst (blue). Staining of select ORFs from one of three independent experiments are shown in A. **B.** As a positive control for PB disassembly, HeLa GFP-Dcp1a cells were transfected with EV or the Kaposi's sarcoma-associated herpesvirus (KSHV) protein, KapB for 48 hours

then fixed and immunostained for KapB (green; Alexa647) or Hedls (PBs; white; Alexa555). Nuclei were stained with Hoechst (blue); Scale bar = 20 μm. **C.** Hedls puncta were quantified as in Fig 1. These data represent three independent replicates (*n = 3*). Statistics were performed using an unpaired Student's *t*-test; (**, *p < 0.01*). **D-E.** DDX6 puncta were quantified using CellProfiler. In D, SARS-CoV-2 ORF-expressing cells were thresholded by Strep-tag II staining intensity. The intensity threshold used was defined as two standard deviations above mean intensity in vector controls. Only DDX6 puncta in cells above this threshold were counted. In E, Strep-tag II staining could not be visualized above this threshold; therefore, DDX6 puncta in all cells were counted (unthresholded). Values are expressed as a fold-change difference normalized to the vector control (hashed line). These data represent three or more independent replicates (*n≥3*). A one-way ANOVA with a Dunnett's post-hoc analysis was performed; bars represent SEM (**, *p < 0.01*). N = nucleocapsid protein, E = envelope protein, M = membrane protein, S = spike protein **F.** HeLa GFP-Dcp1a cells were transfected as in A. 48 hours post transfection cells were lysed. Samples were resolved by SDS-PAGE on 4–15% gradient gels and immunoblotted with a Strep-tag II antibody.

transduced cell population (S6C Fig). Although we detected a strong band of ~50 kDa at the predicted molecular weight for N, we were unable to detect bands for nsp14, ORF3b and ORF7b, while the most prominent band for nsp1 did not migrate at the predicted molecular weight of ~20 kDa (S6 Fig) [99,100]. For these reasons, we decided to focus the remainder of our analysis on the N protein.

As PBs are dynamic RNP granules that undergo transient disassembly and assembly under basal conditions, we wanted to determine whether N-mediated PB loss was caused by enhanced disassembly of PBs or the prevention of PB *de novo* assembly. To determine this, we treated N-expressing HUVECs with sodium arsenite, a known inducer of PB assembly [101] and immunostained PBs for two different PB resident proteins, DDX6 and Hedls. Consistent with our previous observation, significant PB loss was observed after N expression in our untreated control when PBs were labeled using an antibody to DDX6 but not using an antibody to Hedls (Fig 7C–7F). The lack of significant disassembly of Hedls-stained PB puncta by N protein could be attributed to one outlier experiment where the control cells lacked sufficient PBs, a biological variability that is observed between different batches of primary cells such as HUVECs. However, after treatment with sodium arsenite, all N protein-expressing cells displayed robust PB formation regardless of the PB protein antibody used (Fig 7C–7F). This means that N protein expression does not prevent PB formation, but instead triggers PB disassembly, which can be reversed after an appropriate stimulus like sodium arsenite. These data showed that N expression is sufficient to cause PB loss, and that the absence of PBs in N-expressing cells is a result of enhanced PB disassembly not a block in PB formation.

We showed that human CoVs OC43 and 229E also cause PB loss during infection (Fig 3); therefore, we were interested to determine if ectopic expression of nucleocapsid proteins from these or other human CoVs were sufficient to mediate PB disassembly. To test this, we transduced HUVECs with recombinant lentiviruses expressing the N protein from SARS-CoV-2 as well as N protein derived from Betacoronaviruses, MERS-CoV and OC43. Expression of MERS-CoV and OC43 N proteins did not lead to significant PB loss compared to SARS-CoV-2 N (Fig 8A and 8B). We also tested two N proteins from human Alphacoronaviruses, 229E and NL63. 229E N protein failed to induce significant PB loss compared to SARS-CoV-2 N, while unexpectedly, it appeared the expression of NL63 N protein increased PB numbers (Fig 8C and 8D). The significance of this increase is not yet clear. We also expressed the N protein from the more closely related Betacoronavirus, SARS-CoV-1, and observed SARS-CoV-1 N protein caused moderate PB loss, yet SARS-CoV-1 N protein-induced PB disassembly was not deemed significant by statistical analysis (Fig 8E and 8F). We collected protein lysates in parallel and performed immunoblotting to detect each CoV N protein (Fig 8G–8I). We noted that CoV N proteins were not expressed equally (S7 Fig), with the important caveat that a direct comparison between individual N protein expression is not accurate as blotting was performed with different antibodies. Nonetheless, we cannot fully discount expression level as the reason

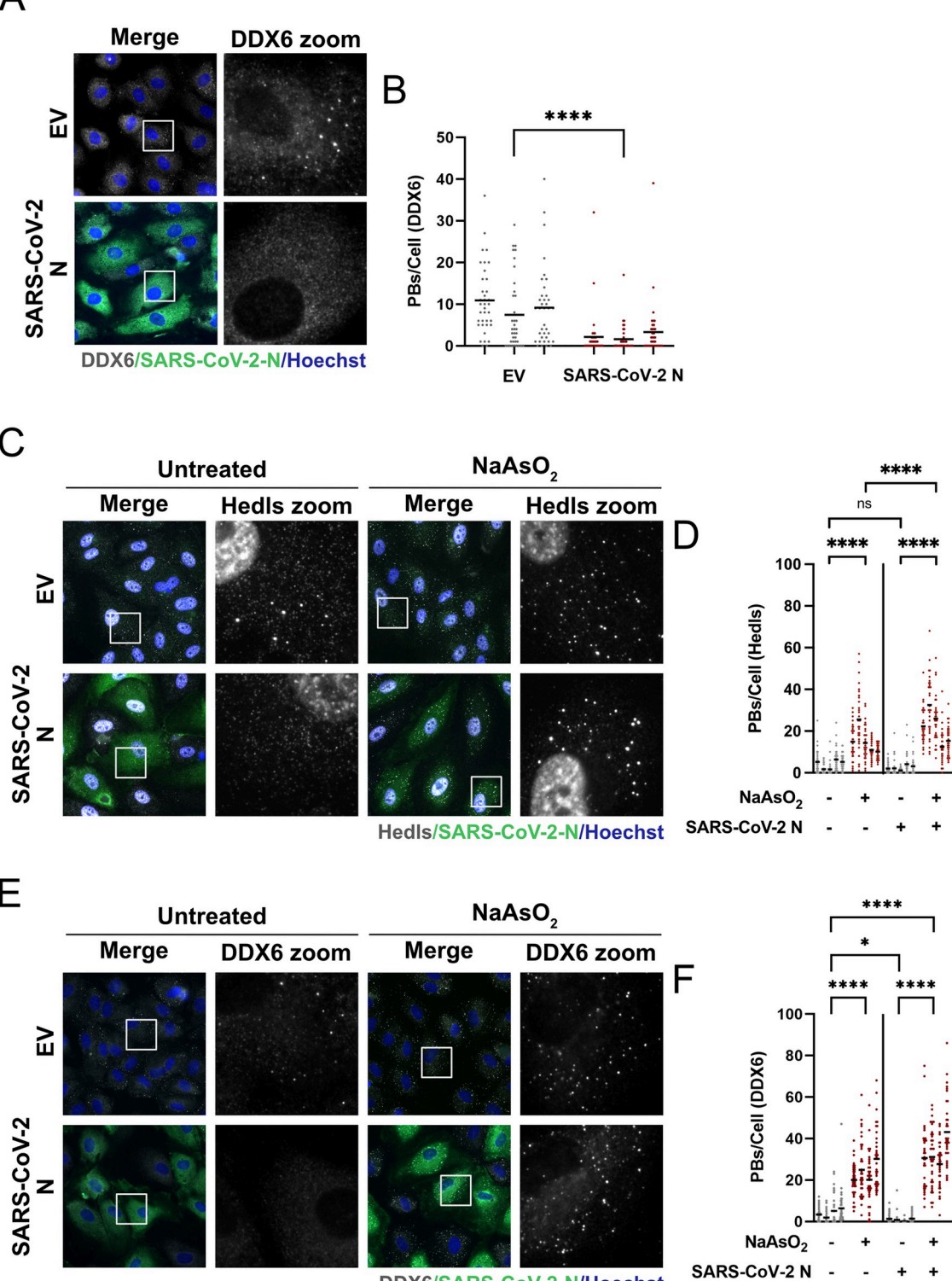

**Fig 7. Ectopic expression of SARS-CoV-2 N elicits processing body disassembly. A-B.** HUVECs were transduced with recombinant lentiviruses ectopically expressing 2X Strep-tagged SARS-CoV-2 N or empty vector control (EV) and selected with puromycin for 48 hours. Samples were fixed and immunostained for Strep-tag II (N; green) and DDX6 (PBs; white). Nuclei were stained with Hoechst (blue). Representative images from one of three independent experiments are shown (*n = 3*). Scale bar = 20 µm. DDX6 puncta were quantified per field of view using CellProfiler as in Fig 1. >30 cells were measured per condition (EV and N) per

replicate. Bars represent the mean. Mann-Whitney rank-sum test was performed (****, $p < 0.0001$). **C-F.** HUVECs were transduced and selected as in A, then treated with 0.25 mM sodium arsenite ($NaAsO_2$) or a vehicle control (Untreated) for 30 minutes, fixed and immunostained for either Hedls (C, white; Alexa555) or DDX6 (D, white; Alexa555) and SARS-CoV-2 N (green; Alexa488). Nuclei were stained with Hoechst (blue). Scale bar = 20 μm. Representative images from one independent experiment of three are shown. These data represent five (C, D) or four (E, F) independent replicates, with >30 cells measured per condition per replicate. Hedls puncta (D) and DDX6 puncta (F) were quantified as in Fig 1. Bars represent the mean. Statistics were performed using a two-way ANOVA with a Tukey's multiple comparisons test (*, $p < 0.0332$; ****, $p < 0.0001$; ns, nonsignificant).

for the discrepancy in PB disassembly. This may be especially important for our analysis of SARS-CoV-1 N protein, which showed an intermediate but non-significant PB disassembly phenotype but was expressed at a lower level that SARS-CoV-2 N protein (Fig 8I). Immuno-blotting of steady state levels of PB resident proteins after N protein overexpression showed that most PB proteins tested remained unchanged (Figs 8G–8I and S7A). The exception to this was the decapping factor, Hedls/EDC4, which was decreased after expression of SARS-CoV-2 N in one set of experiments but increased after expression of N proteins from MERS and OC43 (Figs 8G–8I and S7A). The significance of this observation is also unclear.

To understand if PB disassembly correlates with changes to PB-regulated inflammatory cytokine transcripts in our system, we performed immunofluorescence-RNA fluorescent *in situ* hybridization (IF-FISH) to confirm the localization of inflammatory RNAs to PB foci. RNA FISH was performed for two PB-regulated cytokine transcripts that contain AREs, those encoding IL-6 and TNF, and for the GAPDH RNA, which does not contain AREs and is not expected to localize to PBs [3]. To achieve a better signal-to-noise ratio for detection, we used TNF to induce the transcription of IL-6 and TNF in untreated cells. First, we confirmed that TNF treatment alone did not significantly alter PB dynamics (S8 Fig). We then stained PBs using our antibody to Hedls and co-stained with probes that bound GAPDH, TNF and IL-6 RNA transcripts. We repeatedly observed co-localization of IL-6 and TNF RNA with PBs while in contrast, we observed extremely limited GAPDH co-localization with PBs (Fig 9A). IL-6 and TNF transcripts were also present at much lower levels than GAPDH RNA, consistent with the notion that ARE-containing transcripts are kept low by tight transcriptional control and constitutive decay in PBs [14]. We then performed IF-FISH on SARS-CoV-2 N protein-expressing cells or controls using the same protocol (Figs 9B, 9C, and S9A–S9B). N-expressing cells demonstrated PB disassembly compared to control cells, a phenotype that correlated with the redistribution of the IL-6 and TNF probe signal to the cytoplasm (S9 Fig). We quantified the change in FISH probe signal intensity per cell (Fig 9C). Compared to controls, N protein-expressing cells displayed significantly increased TNF probe signal intensity and a noticeable but non-significant increase in IL-6 probe signal intensity, whereas GAPDH probe signal intensity was not increased (Fig 9C). It is probable that this increase in TNF and IL-6, but not GAPDH, probe signal intensity resulted from the release of these ARE-containing RNAs following N protein-mediated PB disassembly (Figs 9 and S9). These data support our hypothesis that upon PB disassembly, PB-regulated transcripts like TNF are redistributed from PB foci to the cytoplasm where they are available for translation.

We then asked if N protein from SARS-CoV-2 or other human CoVs would increase ARE RNA-containing transcripts by RT-qPCR. We analyzed IL-6 and TNF transcript level, as well as two other ARE-containing RNAs, encoding CXCL8 and COX-2 which were elevated after infection with SARS-CoV-2 and OC43 (Fig 5). In uninduced ECs, the transcription of these mRNAs is minimal; for example, TNF mRNA could not be readily detected by RT-qPCR without transcriptional activation. Therefore, we treated control and N-expressing HUVECs with TNF to activate cytokine transcription which enabled us to assess if N protein expression enhanced cytokine mRNA level post-transcriptionally. In the absence of TNF, no change of

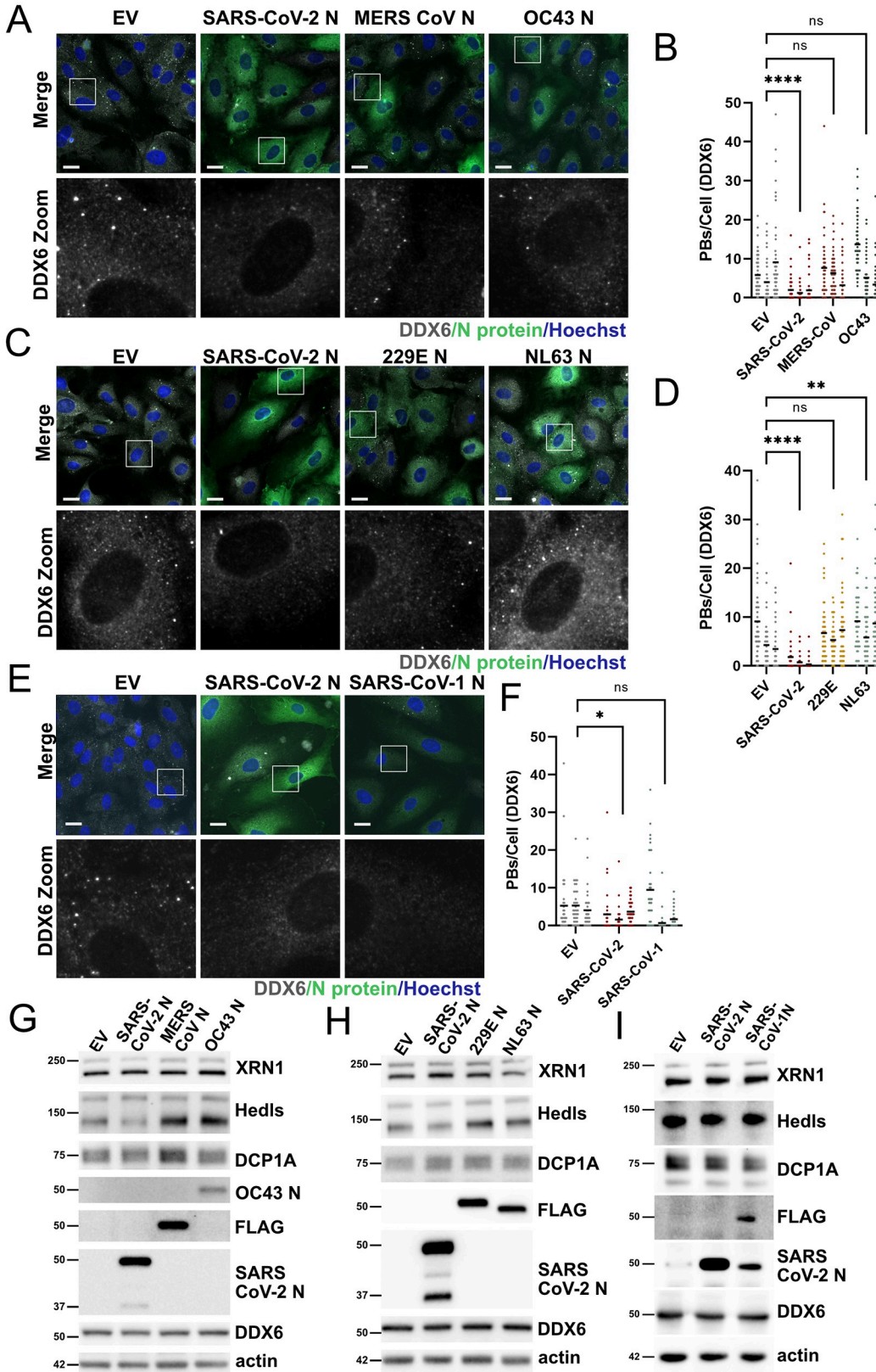

**Fig 8. Processing body disassembly is not a common feature of all human coronavirus N proteins. A-D.** HUVECs were transduced with recombinant lentiviruses ectopically expressing N protein from the Betacoronaviruses MERS-CoV and OC43 (A-B) or N protein from Alphacoronaviruses 229E and NL63 (C-D) or N protein from SARS-CoV-1 (E-F). A control lentiviral expressing an empty vector (EV) was used as a negative control and SARS-CoV-2 N protein expressing lentiviruses were used as a positive control in each experiment. Cells were selected, fixed and immunostained for DDX6 (PBs; white; Alexa555) and either authentic N protein or a FLAG tag (green; Alexa488). Nuclei were stained with Hoechst (blue). Scale bar = 20 μm. DDX6 puncta in EV or N-transduced cells were quantified using CellProfiler as in Fig 1. DDX6 puncta were quantified as in Fig 1. Representative images from one independent experiment of three are shown. These data represent three independent biological replicates ($n = 3$) with >30 cells measured per condition (EV and N) per replicate. Each EV and N replicate pair plotted independently; mean. Statistics were performed using Kruskal-Wallis *H* test with Dunn's correction (*, $p < 0.0332$; **, $p < 0.0021$; ****, $p < 0.0001$; ns, nonsignificant). **G-I.** HUVECs were transduced as above, protein lysate was harvested and immunoblotting was performed using XRN1, Hedls, DCP1A, DDX6, N protein or FLAG, and β-actin specific antibodies. One representative experiment of three is shown.

mRNA abundance was observed for any of the CoV N proteins tested (Fig 10A–10C, 0 hour no treatment). Ectopic expression of SARS-CoV-2 N enhanced transcript levels of IL-6, CXCL8, and TNF 24 hours after transcription was induced; however, this enhancement was not deemed significant by statistical analysis (Fig 10A–10C). In contrast, we did not observe any increase in transcript levels after expression of N protein from MERS-CoV, OC43, 229E, or NL63 (Fig 10A–10C), consistent with earlier observations that their expression does not induce PB loss.

Taken together, we present the finding that three human CoVs induce PB loss and that for SARS-CoV-2, the N protein is responsible for this loss. Moreover, we tested five other human CoV N proteins and found that of these, only SARS-CoV-2 N was sufficient to induce significant PB disassembly and concomitantly redistributed and enhanced levels of PB-localized cytokine mRNAs.

## Discussion

In this manuscript, we present the first evidence to show that human CoVs, including SARS-CoV-2, induce PB loss after infection. PBs are fundamental sites of post-transcriptional control of gene expression and are particularly relevant to the regulation of cytokine production. Our major findings are as follows: i) Three human CoVs, SARS-CoV-2, OC43, and 229E induced PB loss. ii) The SARS-CoV-2 nucleocapsid (N) protein was sufficient to cause PB loss. iii) N protein expression correlated with the elevated abundance and re-distribution of the PB-regulated cytokine transcript encoding TNF. Taken together, these data point to PB loss as a feature of SARS-CoV-2 infection. Moreover, because virus-induced PB disassembly elevates some PB-regulated cytokine transcripts, this phenotype may contribute to the dysregulation of proinflammatory molecules observed in severe SARS-CoV-2 infection.

We screened 27 SARS-CoV-2 gene products by transfection in HeLa cells [96] and initially identified eight candidate genes that reduced PB numbers (Fig 6). Validation of a subset of these in HUVECs revealed that the most robust and consistent mediator of PB loss was the SARS-CoV-2 N protein (Fig 7). Since the submission of this work, one additional study showed that SARS-CoV-2 N protein caused PB loss, further strengthening our findings [102].

The N protein is the most abundantly produced protein during CoV replication [103]. The SARS-CoV-2 N protein is 419 amino acids long and has two globular and three intrinsically disordered protein domains, including a central disordered serine-arginine (SR-rich) linker region [103–105]. The N protein is a multifunctional RNA-binding protein (RBP) essential for viral replication; it coats the viral genome and promotes viral particle assembly [46,105–108]. Several recent reports have shown that N protein undergoes liquid-liquid phase separation with RNA, an event which may be regulated by phosphorylation of multiple serine residues in the SR-region and is an important feature of viral assembly [108–113]. The N protein is also an

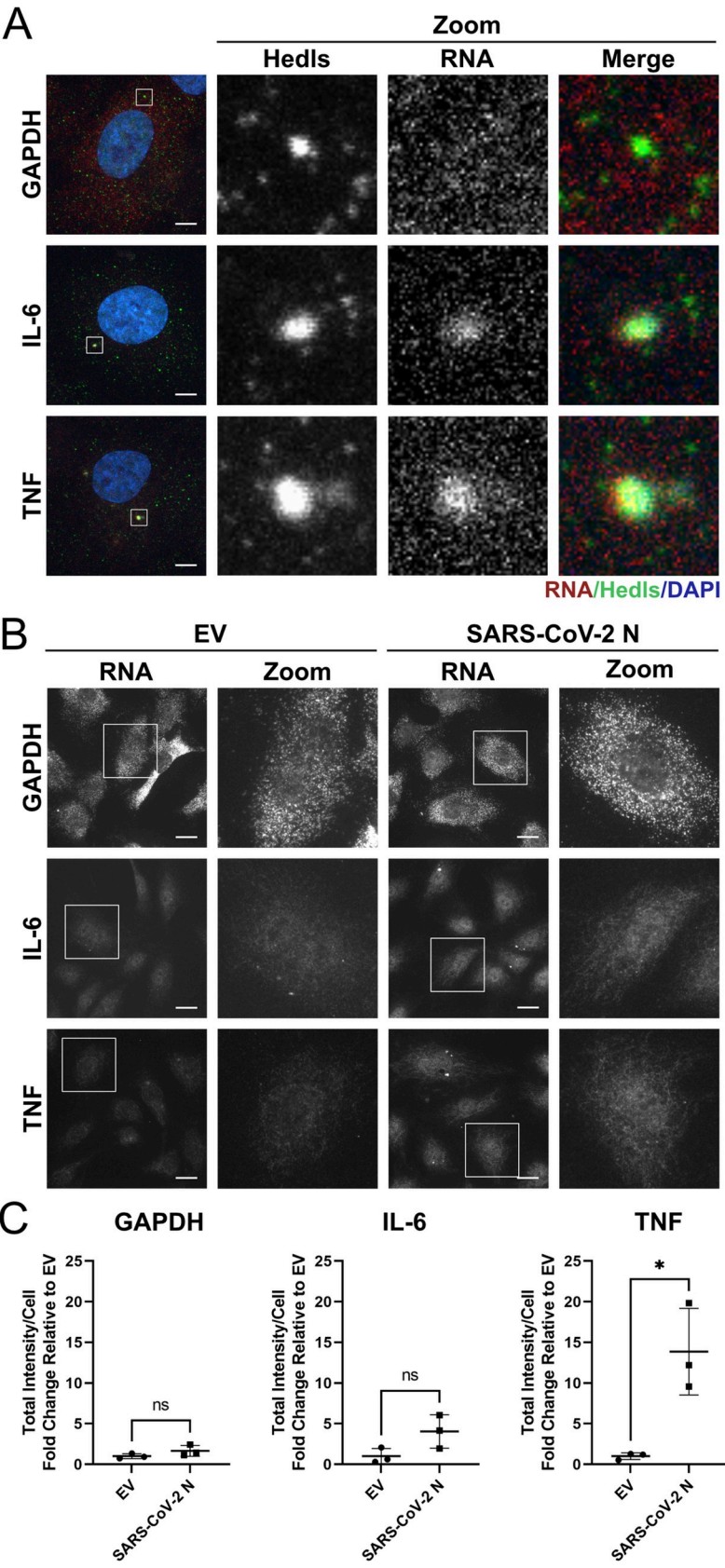

**Fig 9. ARE-mRNAs colocalize with processing bodies. A.** HUVECs were treated with 0.01 ng/L soluble TNF to increase transcription of ARE-containing cellular mRNAs for 24 hours. Cells were fixed and immunostained for Hedls (green; Alexa488) prior to hybridization with Stellaris probes specific for GAPDH, IL-6 and TNF. Nuclei were stained with DAPI. Cells were imaged using a Zeiss LSM 880 confocal microscope with Airyscan using the 63X objective. At least three Z-stacks were acquired per condition (probe) per replicate and a maximal intensity projection (MIP) image is presented. One representative experiment of three is shown. Scale bar = 5 μm. **B-C.** HUVECs were transduced with recombinant lentiviruses ectopically expressing SARS-CoV-2 N or an EV control. Cells were selected, fixed and immunostained for Hedls (Alexa488, not shown here) prior to hybridization with Stellaris probes specific for GAPDH, IL-6 and TNF (Quasar 670). Nuclei were stained with DAPI. Only probe-specific staining is displayed for simplicity. Scale bar = 20 μm. Representative images from one of three independent experiments are shown. Total signal intensity per cell was quantified using CellProfiler and is represented as fold-change relative to the intensity in the EV control. Statistics were performed using an unpaired Student's $t$-test. $n = 3$; mean ± SD (*, $p < 0.05$; ns, nonsignificant).

important modulator of antiviral responses [85,114,115]. A recent study showed that low doses of N protein suppressed the production of IFN and some PB-regulated inflammatory cytokines, while high doses of N protein promoted their production [85]. These observations are consistent with our phenotype of PB disassembly, which correlates with later infection times, high expression of N protein and immunofluorescent staining throughout the cytoplasm (Figs 1 and 7).

We subsequently screened five other coronavirus N proteins from OC43, MERS, 229E, NL63, and SARS-CoV-1 and discovered that the phenotype of N-mediated PB disassembly was not conserved among N proteins but was unique to SARS-CoV-2 N, and perhaps SARS-CoV-1 N protein which appeared to disassemble PBs in two of three independent experiments (Fig 8). Despite conservation of some structural motifs, the N protein contains several regions of intrinsic disorder, a feature that may explain why human CoV N proteins display low sequence conservation at the amino acid level (~25–50%) and have been reported to exhibit different properties [116]. In contrast, SARS-CoV-1 and SARS-CoV-2 N proteins have ~94% amino acid identity [117], making it difficult to reconcile our observation that PB disassembly induced by SARS-CoV-2 N protein was significant, while PB disassembly induced by SARS-CoV-1 N protein was not (Fig 8). One difference that we observed by immunoblotting was the presence of a lower molecular weight band recognized by our anti-N antibody for SARS-CoV-2. We did not observe any smaller N protein products after OC43 infection, transduction with OC43 N protein or transduction with C-terminally Flag-tagged N proteins from 229E, NL63, MERS-CoV or SARS-CoV-1 (Figs 3 and 8). Steady state levels of the lower molecular weight N product increased over the course of SARS-CoV-2 infection, consistent with the timing of PB disassembly. Other groups have noted that the SARS-CoV-2 variant of concern (VOC), Alpha, produces an additional subgenomic mRNA from which a truncated version of N, termed N*, can be produced [68,118,119]. Translation of the N* ORF is predicted to start at an internal in-frame methionine residue (Met210) within the N protein [118,119]. Alignment of the SARS-CoV-2 N protein sequence against other N proteins revealed that of the human CoVs, only SARS-CoV-1 and SARS-CoV-2 N retained a methionine at position 210 (229E: NC_002645.1, HKU1: NC_006577.2, MERS: NC_019843.3, NL63: NC_005831.2, OC43: NC_006213.1, SARS-1: NC_004718.3, SARS-2: NC_045512.2). However, a lower molecular weight N protein product was not observed on our immunoblots for SARS-CoV-1 N protein, leading us to speculate that the downstream Met residue may not be used for translation initiation in this case. Viruses capitalize on downstream methionine residues to translate truncated protein products with subcellular localization or functions that differ from their full-length counterparts as a clever way to increase coding capacity [120,121]. Our ongoing investigation of the precise nature of the SARS-CoV-2 N protein truncation product we observe during infection and overexpression may reveal that it has a specific role in PB disassembly.

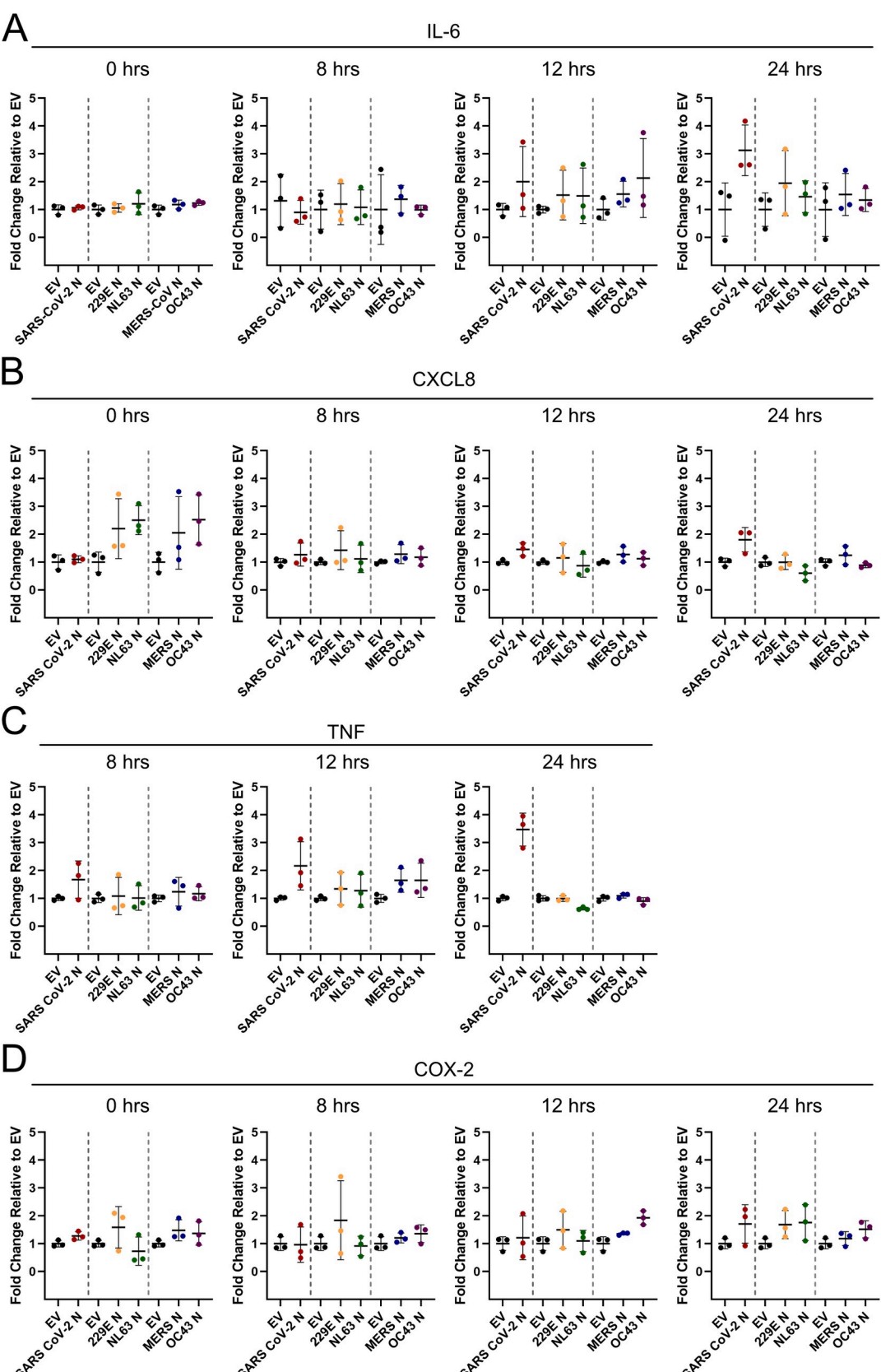

**Fig 10. Ectopic expression of SARS-CoV-2 N and analysis of selected ARE-mRNAs. A-C.** HUVECs were transduced with recombinant lentiviruses ectopically expressing Alpha- and Betacoronavirus N proteins or empty vector (EV) control, selected, and treated with 0.01 ng/L soluble TNF to increase transcription of ARE-containing cellular mRNAs. Total RNA was harvested at 0, 8, 12, and 24 hours post TNF treatment and RT-qPCR was performed using IL-6 (A), CXCL8 (B), TNF (C) and HPRT (house-keeping gene) specific primers. Values are represented as fold-change relative to EV-transduced cells for each time point. These data represent three independent experiments ($n = 3$); mean ± SD.

The PB protein MOV10, and other components of RNA processing machinery, were revealed as potential interactors with the N protein [102,122]; however, we did not observe colocalization of N protein with PBs after immunofluorescent staining of SARS-CoV-2 infected cells or N-expressing cells. Based on our data, we consider two possible mechanisms of N protein mediated PB disassembly. First, N protein may mediate PB disassembly by phase separation with a PB protein(s). This is similar to what has already been shown for N-mediated disruption of cytoplasmic stress granules, important cytoplasmic biomolecular condensates that correlate with cellular translational shutdown [35]. N protein localizes to stress granules and binds the essential protein, G3BP1, preventing its interaction with other stress granule proteins and blocking stress granule formation [123–125]. Although the precise domain required for this effect has been debated, more than one report suggests that the N-terminal intrinsically disordered region is required for stress granule disruption [124,126]. Second, a possible reason for PB loss may be the indiscriminate binding of RNA by N protein. N protein could be acting as sponge for RNA, pulling it out of cytoplasm, thereby reducing the RNA-protein interactions required for phase separation of PBs [24,123]. We are currently engaged in site-directed and truncation mutagenesis studies to determine the precise region(s) of SARS-CoV-2- N that is essential for PB disassembly.

Prior to this report, little was known about CoVs and PBs, and the information that was published was contradictory. Infection with murine hepatitis virus (MHV) was reported to increase PBs, whereas transmissible gastroenteritis coronavirus (TGEV) decreased PBs [86,87]. Since the initiation of our study, one additional publication used ectopic expression of one of the SARS-CoV-2 CoV proteases, nsp5, to test if it was capable of PB disassembly. Consistent with the results of our screen (Fig 6), nsp5 did not mediate PB loss [32]. In this manuscript, we now confirm that SARS-CoV-2, OC43, and 229E induce PB disassembly (Figs 1–3) [34]. We also observed that different human CoVs cause PB loss using different viral gene products; SARS-CoV-2 utilizes N protein, however OC43 and 229E do not, a diversity which further underscores that convergence on PB disassembly by viruses likely benefits viral replication. Because PBs are composed of numerous cellular molecules with established (e.g. APOBEC, MOV10) or potential (decapping enzymes and exonucleases that could degrade viral RNA) antiviral activities, it is possible that viruses may target PBs for disassembly to negate their antiviral activity [28,29,31,39–45,127]. A recent screen for cellular proteins that bind SARS-CoV-2 viral RNA captured two PB proteins (Lsm14a and MOV10), which suggests CoV RNA may be shuttled to PBs [128]. That said, our evidence does not yet discern if the proposed antiviral role of PB-localized enzymes is promoted by phase separation of molecules into PBs or not; if so, we would predict that the antiviral function of these molecules is lost when PB granules decondense. Emerging evidence suggests that activity of the decapping enzymatic complex is increased by phase separation and decreased in solution [9,24,129]. Thus, we speculate that PBs are direct-acting antiviral granules that can restrict virus infection when present as visible condensates; for this reason, they are targeted for disassembly by most viruses.

One possibility is that PBs are antiviral because their proteins help the cell respond to signals that activate innate immune pathways [28,30,130,131]. In support of this, TRAF6 was shown to control Dcp1a localization to PBs using ubiquitylation, suggesting that antiviral

signaling is more complex than previously appreciated and integrates transcriptional responses with cytokine mRNA suppression in PBs [130,132]. Moreover, the PB protein Lsm14A has been shown to bind to viral RNA/DNA after infection-induced PB disassembly to promote IRF3 activation and IFN-β production [131]. Although it remains unclear if the higher order condensation of many proteins into PBs is required for their proposed antiviral activity, what is clear is that the outcome of PB disassembly is a reversal of the constitutive decay or translational suppression of cytokine mRNA that would normally occur there [7,12–15,133]. Our data support a role for PBs as sensors of virus infection that release cytokine transcripts from repression when they disassemble. Using IF-FISH, we observed that RNAs encoding IL-6 and TNF, two molecules elevated in severe SARS-CoV-2 disease, localized to PBs. Upon SARS-CoV-2 N-induced PB disassembly, these transcripts re-localized from PB foci to the cytoplasm, a redistribution that was also accompanied by an increase in FISH probe signal intensity signifying increased RNA abundance for the TNF transcript (Fig 9). Although we did not see a statistically significant increase in IL-6 and TNF RNA in SARS-CoV-2 N-expressing cells using a population-based assay like RT-qPCR (Fig 10), our single-cell analysis by IF-FISH suggests that the biological relevance of our observation derives from the combined increase of transcript abundance plus the redistribution of cytokine transcripts from PB foci to the cytoplasm. We speculate that viral infection causes PB loss and this event is viewed as a danger signal by the cell: it relieves cytokine mRNA suppression and re-localizes these transcripts, returning them to the cytoplasmic pool of RNA to permit translation of proinflammatory cytokines that then act as a call for reinforcements. In this way, PB disassembly is connected to the innate immune response and is one of many signals that notify the immune system that a cell is infected. In situations where interferon responses are delayed or defective, as is emerging for SARS-CoV-2 and severe COVID-19 [65–67], PB disassembly may occur to alert the immune system of an infection, and may be a contributing factor to pathogenic cytokine responses.

In summary, our work adds to a growing body of literature which describes that many viruses target PBs for disassembly, supporting the idea that PBs restrict viral infection. We showed that the N protein of SARS-CoV-2 is sufficient for PB disassembly, and this phenotype correlated with elevated levels of PB-localized cytokine transcripts which are elevated during SARS-CoV-2 infection [53]. Not only does this work describe a previously uncharacterized cellular event that occurs during CoV infection, but we have identified a novel mechanism which may contribute to the dysregulated cytokine response exhibited by severe SARS-CoV-2 infection.

## Materials and methods

### Cell culture and drug treatments

All cells were maintained in humidified 37˚C incubators with 5% $CO_2$ and 20% $O_2$. Vero E6 (ATCC), HEK293T cells (ATCC), HeLa Tet-Off cells (Clontech) and HeLa Flp-In TREx GFP-Dcp1a cells (a generous gift from Dr. Anne-Claude Gingras) were cultured in DMEM (Thermo Fisher) supplemented with 100 U/mL penicillin, 100 μg/mL streptomycin, 2 mM L-glutamine (Thermo Fisher) and 10% heat-inactivated fetal bovine serum (FBS, Thermo Fisher). Calu3 (ATCC) and MRC-5 cells (ATCC, a generous gift from Dr. David Proud) were cultured in EMEM (ATCC) supplemented with 100 U/mL penicillin, 100 μg/mL streptomycin, 2 mM L-glutamine and 10% FBS (Thermo Fisher). HUVECs (Lonza) were cultured in endothelial cell growth medium (EGM-2) (Lonza). HUVECs were seeded onto tissue culture plates or glass coverslips coated with 0.1% (w/v) porcine gelatin (Sigma) in 1x PBS. For sodium arsenite treatment, HUVECs were treated with 0.25 mM sodium arsenite (Sigma) or a vehicle control for 30 min.

## Plasmids and cloning

pLenti-IRES-Puro SARS-CoV-2 plasmids were a generous gift from the Krogan Lab [96]. pLJM1-OC43-N was cloned from pGBW-m4134906, a gift from Ginkgo Bioworks & Benjie Chen (Table 1; Addgene plasmid #151960; http://n2t.net/addgene:151960; RRID: Addgene_151960) using BamHI and EcoRI restriction sites (NEB). pLJM1-NL63-N-FLAG was cloned from pGBW-m4134910, a gift from Ginkgo Bioworks & Benjie Chen (Addgene plasmid #151939; http://n2t.net/addgene:151939; RRID:Addgene151939) using BamHI and EcoRI. pLJM1-229E-N-FLAG was cloned from pGBW-m4134902, a gift from Ginkgo Bioworks & Benjie Chen (Addgene plasmid #151912; http://n2t.net/addgene:151912; RRID: Addgene151912) using BamHI and EcoRI. Codon-optimized pLJM1-MERS-CoV-N-FLAG was cloned from SinoBiological (cat #VG40068-CF) using BamHI and EcoRI. Codon-optimized pLJM1-SARS-CoV-N-FLAG was cloned SinoBiological (cat# VG40588-NF) using BamHI and EcoRI. The Kaposi's sarcoma-associated herpesvirus Kaposin B (KapB, KS lung isolate) clone in pLJM1 was previously described [91].

## Transient transfections

Transient transfections were performed using Fugene (Promega) according to manufacturer's guidelines. Briefly, HeLa Flp-In TREx GFP-Dcp1a cells were seeded in 12-well plates at 150,000 cells/well in antibiotic-free DMEM containing FBS. Cells were transfected with 1 μg of DNA and 3 μL of Fugene for 48 hours before processing.

## Production and use of recombinant lentiviruses

All recombinant lentiviruses were generated using a second-generation system. HEK293T cells were transfected with psPAX2, MD2-G, and the lentiviral transfer plasmid containing a gene of interest using polyethylenimine (PEI, Polysciences). 6 hours after transfection, serum-free media was replaced with DMEM containing serum but no antibiotics. Viral supernatants were harvested 48 hours post transfection and frozen at -80°C until use. For transduction, lentiviruses were thawed at 37°C and added to target cells in complete media containing 5 μg/mL polybrene (Sigma). After 24 hours, the media was replaced with selection media containing 1 μg/mL puromycin or 5 μg/mL blasticidin (ThermoFisher) and cells were selected for 48 h before proceeding with experiments.

## Immunofluorescence

Cells were seeded onto 18mm round, #1.5 coverslips (Electron Microscopy Sciences) for immunofluorescence experiments. Following treatment, cells were fixed for 10 or 30 (if infected with SARS-CoV-2) minutes in 4% (v/v) paraformaldehyde (Electron Microscopy Sciences). Samples were permeabilized with 0.1% (v/v) Triton X-100 (Sigma-Aldrich) for 10 minutes at room temperature and blocked in 1% human AB serum (Sigma-Aldrich) in 1X PBS 1 hour at room temperature. Primary and secondary antibodies (Table 2) were diluted in 1% human AB serum and used at the concentrations in Table 2. In all figures except Figs 9 and S9, PBs were labeled with an Alexa 555-conjugated secondary antibody to minimize bleed through into the PB channel that was being quantified. Nuclei were stained with 1 μg/ml Hoechst (Invitrogen). Samples were mounted with Prolong Gold AntiFade mounting media (Thermo-Fisher). Images were captured using a Zeiss AxioObserver Z1 microscope with the 40X oil-emersion objective unless otherwise stated in the respective figure caption. To account for variability in staining, all experiments contained internal controls (negative control; mock or EV).

**Table 1. Plasmids.**

| Plasmid | Use | Source | Mammalian Selection |
|---|---|---|---|
| pLJM1-Puro | Empty Vector Control | [91] [136] | Puromycin |
| pLJM1-BSD | Empty Vector Control | [136] | Blasticidin |
| pLJM1-ACE2 | Overexpression | | Blasticidin |
| pMD2.G | Lentivirus Generation | Addgene #12259 | N/A |
| psPAX2 | Lentivirus Generation | Addgene #12260 | N/A |
| pLJM1-OC43-N | Overexpression | Addgene #151960 | Puromycin |
| pLJM1-229E-N-FLAG | Overexpression | Addgene #151912 | Puromycin |
| pLJM1-NL63-N-FLAG | Overexpression | Addgene #151939 | Puromycin |
| pLJM1-MERS-CoV-N-FLAG | Overexpression | SinoBiological #VG40068-CF | Puromycin |
| pLJM1-SARS-CoV-1-N-FLAG | Overexpression | SinoBiological # VG40588-NF | Puromycin |
| pLenti-SARS-CoV-2-IRES-strep | Overexpression library | N Krogan (UCSF) [96] | Puromycin |

During image acquisition, exposure time was kept consistent within an independent experiment.

## Immunofluorescence (IF) RNA Fluorescent *In Situ* Hybridization (FISH)

IF-FISH was performed according to manufacturer's protocol (Stellaris IF-FISH protocol). Briefly, cells were fixed in 4% formaldehyde (Sigma-Aldrich) for 10 minutes then permeabilized with 0.1% Triton X-100 (Sigma-Aldrich) in PBS for 10 minutes. Primary Hedls antibody was diluted in PBS and incubated for 1 hour at room temp, following which an Alexa 488-conjugated secondary antibody was likewise diluted in PBS and incubated for 1 hour at room temp. In some experiments, N protein was also stained using primary anti-N protein rabbit

**Table 2. Antibodies.**

| Antibody | Species | Vendor/Catalog # | Application | Dilution |
|---|---|---|---|---|
| Strep-Tag II | Mouse | Sigma (71590-M) | Immunofluorescence<br>Immunoblot | 1:1000<br>1:1000 |
| DDX6 | Rabbit | Bethyl (A300-461) | Immunofluorescence<br>Immunoblot | 1:1000<br>1:1000 |
| Hedls | Mouse | Santa Cruz (sc-8418) | Immunofluorescence<br>Immunoblot | 1:1000<br>1:1000 |
| Dcp1a | Rabbit | Novus (H00055802-M06) | Immunoblot | 1:1000 |
| Xrn1 | Mouse | Abcam (ab231197) | Immunoblot | 1:1000 |
| Coronavirus OC43 Nucleocapsid | Mouse | Millipore (MAB-9012) | Immunofluorescence<br>Immunoblot | 1:500<br>1:1000 |
| SARS-CoV-2 Nucleocapsid | Rabbit | Novus (NBP3-05730) | Immunofluorescence | 1:1000 |
| SARS-CoV-2 Nucleocapsid | Mouse | Novus (NBP3-05706) | Immunofluorescence<br>Immunoblot | 1:1000<br>1:1000 |
| dsRNA clone J2 | Mouse | Millipore (MABE-1134) | Immunofluorescence | 1:100 |
| FLAG | Mouse | CST (8146) | Immunofluorescence<br>Immunoblot | 1:1000<br>1:1000 |
| Actin HRP-conjugated | Rabbit | CST (5215) | Immunoblot | 1:10,000 |
| Alexa Fluor 555 anti-mouse IgG | Donkey | Invitrogen (A31570) | Immunofluorescence | 1:1000 |
| Alexa Fluor 555 anti-rabbit IgG | Donkey | Invitrogen (A31572) | Immunofluorescence | 1:1000 |
| Alexa Fluor 488 anti-mouse IgG | Goat | Invitrogen (A11029) | Immunofluorescence | 1:1000 |
| Alexa Fluor 488 anti-rabbit IgG | Chicken | Invitrogen (A21441) | Immunofluorescence | 1:1000 |
| Alexa Fluor 647 anti-mouse IgG | Chicken | Invitrogen (A21463 | Immunofluorescence | 1:1000 |

**Table 3. Probes for RNA Fluorescent in situ hybridization.**

| Target: | IL-6 | TNF |
|---|---|---|
| | gatagagcttctctttcgtt | cgtctgagggttgttttcag |
| | agaaggagttcatagctggg | gggtcagtatgtgagaggaa |
| | agaaggcaactggaccgaag | atcatgctttcagtgctcat |
| | cggctacatctttggaatct | caaagtgcagcaggcagaag |
| | aagaggtgagtggctgtctg | cggggttcgagaagatgatc |
| | atttgtttgtcaattcgttc | cttgagggtttgctacaaca |
| | agatgccgtcgaggatgtac | accagctggttatctctcag |
| | tcacacatgttactcttgtt | ctgatggtgtgggtgaggag |
| | tctttggaaggttcaggttg | tctggtaggagacggcgatg |
| | aagcatccatctttttcagc | ctcttgatggcagagaggag |
| | ctcctcattgaatccagatt | gatagatgggctcataccag |
| | tgatgattttcaccaggcaa | attgatctcagcgctgagtc |
| | ctggaggtactctaggtata | caaagtcgagatagtcgggc |
| | tcctcactactctcaaatct | aaagtagacctgcccagact |
| | tactcatctgcacagctctg | ctcctcacagggcaatgatc |
| | caggaactggatcaggactt | ttgggaaggttggatgttcg |
| | gggtggttattgcatctaga | gtctgaaggaggggggtaata |
| | tgagatgagttgtcatgtcc | gtggtcttgttgcttaaagt |
| | aactccttaaagctgcgcag | ctgaatcccaggtttcgaag |
| | catgctacatttgccgaaga | ttgaattcttagtggttgcc |
| | acaggtttctgaccagaaga | agggatcaaagctgtaggcc |
| | aacataagttctgtgcccag | tggtctccagattccagatg |
| | ctcatacttttagttctcca | cattctggccagaaccaaag |
| | ttcaaactgcatagccactt | taggtgaggtcttctcaagt |
| | ccaagaaatgatctggctct | aaggtccacttgtgtcaatt |
| | atttgaggtaagcctacact | acatctggagagaggaaggc |
| | | | cgtgtctcaaggaagtctgg |
| | | ctacatgggaacagcctatt |
| | | caaaagaaggcacagaggcc |
| | | ttggtcaccaaatcagcatt |
| | | agaggctcagcaatgagtga |
| | | gggcgattacagacacaact |
| | | ctttatttctcgccactgaa |

antibody and an Alexa 555-conjugated secondary antibody. After IF, cells were fixed in 4% formaldehyde for 10 minutes, washed with 1X PBS, and incubated in Wash Buffer A (LGC Biosearch) for 5 minutes. Stellaris probes; GAPDH Quasar 670 (LGC Biosearch, VSMF 2151–5, probe sequences proprietary), IL-6 Quasar 670 (LGC Biosearch, SMF 1065–5, custom order, probe sequences detailed in Table 3) and TNF Quasar-670 (LGC Biosearch, SMF 1065–5, custom order, probe sequences detailed in Table 3) were diluted in Hybridization Buffer (LGC Biocearch) to a concentration of 125 nM and cells were incubated for hybridization at 37°C overnight. The following day, cells were washed sequentially with Wash Buffer A for 30 minutes, Wash Buffer A containing 5 ng/mL DAPI (Invitrogen) for 30 minutes, and Wash Buffer B (LGC Biosearch) for 5 minutes. Samples were mounted with Prolong Gold AntiFade mounting media (ThermoFisher). Samples were imaged on a Zeiss LSM 880 laser scanning confocal microscope using the 63X 1.4 NA Oil Plan-Apo objective and the following lasers (excitation wavelengths): Diode (405nm), Argon (488nm), and Helium/Neon (561nm and 633nm) with

emission filter ranges of 410-513nm, 489-605nm, 570-659nm and 638-755nm, respectively. At least three z-stacks were acquired per condition, maximal intensity projections (MIPs) of 18 slices with dimensions of 1912 x 1912 pixels are presented.

## Immunoblotting

Cells were lysed in 2X Laemmli buffer and stored at -20°C until use. The DC Protein Assay (Bio-Rad) was used to quantify protein concentration as per the manufacturer's instructions. 10–15 μg of protein lysate was resolved by SDS-PAGE on TGX Stain-Free acrylamide gels (BioRad). Total protein images were acquired from the PVDF membranes after transfer on the ChemiDoc Touch Imaging system (BioRad). Membranes were blocked in 5% BSA in TBS-T (Tris-buffered saline 0.1% Tween-20). Primary and secondary antibodies were diluted in 2.5% BSA, dilutions can be found in Table 2. Membranes were visualized using Clarity Western ECL substrate and the ChemiDoc Touch Imaging system (BioRad). Densitometry was conducted using ImageJ, area under the curve for each band was calculated, normalized to the respective beta actin loading control band, and presented as fold change.

## Quantitative PCR

RNA was collected using the RNeasy Plus Mini Kit (Qiagen) according to the manufacturer's instructions and stored at -80°C until further use. RNA concentration was determined using NanoDrop One$^C$ (ThermoFisher) and 500 ng of RNA was reverse transcribed using qScript XLT cDNA SuperMix (QuantaBio) using a combination of random hexamer and oligo dT primers, according to the manufacturer's instructions. Depending on starting concentration, cDNA was diluted between 1:10 and 1:20 for qPCR experiments and SsoFast EvaGreen Mastermix (Biorad) was used to amplify cDNA. The ΔΔquantitation cycle (Cq) method was used to determine the fold change in expression of target transcripts using HPRT as a housekeeping control gene. Variance in the mock or empty vector samples was calculated by dividing the ΔCt value of a single replicate by the average ΔCt value of all replicates for that specific gene and condition. qPCR primer sequences can be found in Table 4.

## Virus stocks and virus propagation

Experiments with SARS-CoV-2 and variants were conducted in a containment level-3 (CL3) facility, and all standard operating procedures were approved by the CL3 Oversight Committee and Biosafety Office at the University of Calgary. Stocks of SARS-CoV-2 Toronto-01 isolate (SARS-CoV-2/SB3-TYAGNC) [90], Alpha, Beta, Gamma and Delta were propagated in Vero E6 cells. To produce viral stocks, Vero E6 cells were infected at an MOI of 0.01 for 1 hour in serum-free DMEM at 37°C. Following adsorption, DMEM supplemented with 2% heat inactivated FBS and 100 U/mL penicillin, 100 μg/mL streptomycin, 2 mM L-glutamine was added to the infected wells. 24–60 hours post infection, the supernatant was harvested and centrifuged at 500 x *g* for 5 minutes to remove cellular debris. Virus stocks were aliquoted and stored at -80°C for single use. SARS-CoV-2 titres were enumerated using plaque assays on Vero E6 cells as previously described [134] using equal parts 2.4% w/v semi-solid colloidal cellulose overlay (Sigma; prepared in ddH$_2$O) and 2X DMEM (Wisent) with 1% heat inactivated FBS and 100 U/mL penicillin, 100 μg/mL streptomycin, 2 mM L-glutamine.

Experiments with hCoV-OC43 (ATCC VR-1558) and hCoV-229E (ATCC VR-740) were conducted in under containment level-2 conditions. hCoV-OC43 and hCoV-229E were propagated in Vero E6 and MRC-5 cells, respectively. Cells were infected at an MOI of 0.01 for 1 hour in serum-free media at 33°C. Following adsorption, the viral inoculum was removed and replaced with fresh media supplemented with 2% heat inactivated FBS and 100 U/mL

**Table 4. qPCR primers.**

| Target | Direction | Sequence 5'→3' |
|---|---|---|
| HPRT | Forward | CTTTCCTTGGTCAGGCAGTATAA |
| HPRT | Reverse | AGTCTGGCTTATATCCAACACTTC |
| 18S | Forward | TTCGAACGTCTGCCCTATCAA |
| 18S | Reverse | GATGTGGTAGCCGTTTCTCAGG |
| IL6 | Forward | GAAGCTCTATCTCGCCTCCA |
| IL6 | Reverse | TTTTCTGCCAGTGCCTCTTT |
| CXCL8 | Forward | AAATCTGGCAACCCTAGTCTG |
| CXCL8 | Reverse | GTGAGGTAAGATGGTGGCTAAT |
| IL-1β | Forward | CTCTCACCTCTCCTACTCACTT |
| IL-1β | Reverse | TCAGAATGTGGGAGCGAATG |
| TNF | Forward | TCGAACCCCGAGTGACAA |
| TNF | Reverse | AGCTGCCCCTCAGCTTG |
| GM-CSF | Forward | AAATGTTTGACCTCCAGGAGCC |
| GM-CSF | Reverse | ATCTGGGTTGCACAGGAAGTT |
| COX-2 | Forward | CCCTTGGGTGTCAAAGGTAA |
| COX-2 | Reverse | GCCCTCGCTTATGATCTGTC |
| 229E 5' Leader | Forward | AGTTGCTTTTTAGACTTTGTGTCT |
| 229E gRNA (ORF1a) | Reverse | CAAGTGTCACACGGTTGCAG |
| 229E sgRNA (N) | Reverse | CCACGTTGTGGTTCAGATGC |

penicillin, 100 µg/mL streptomycin, 2 mM L-glutamine. After 5–6 days post infection, the supernatant was harvested and cellular debris was cleared by centrifugation. Virus stocks were aliquoted and stored at -80˚C. hCoV-OC43 and hCoV-229E titres were enumerated using Reed and Muench tissue-culture infectious dose 50% ($TCID_{50}$) in Vero E6 or MRC-5 cells, respectively.

## Virus infection

For experimental infections, cells were seeded into wells to achieve ~80% confluency at the time of infection. The growth media was removed and replaced with 100 µL of viral inoculum diluted in serum-free DMEM to reach the desired MOI and incubated at 37˚C for 1 hour, rocking the plate every 10 minutes. Following incubation, the virus inoculum was removed and replaced with 1 mL of complete growth media.

## Processing body and FISH probe intensity quantification

Processing bodies were quantified using an unbiased image analysis pipeline generated in the freeware CellProfiler4.0.6 (cellprofiler.org) [135] as in [91]. First, nuclear staining was used to identify individual cells applying a binary threshold and executing primary object detection between 65 and 200 pixels for HUVECs/HUVEC$^{ACE2}$ and 40 and 200 pixels for Calu3s. For each identified object (nuclei), the peripheral boundary of each cell was defined using the "Propagation" function. Propagation distance was customized depending on cell type to account for variance in cell size (150 pixel radius from the nuclei for HUVECs/HUVEC$^{ACE2}$, 60 pixel radius from nuclei for Calu3s). Using a subtractive function to remove the nuclei from the total cell area, the cytoplasm of each cell was defined. The cytoplasm area mask was then applied to the matched image stained for PB proteins (DDX6 or Hedls) to count only cytoplasmic puncta. Importantly, multiple nuclei in close proximity (e.g., within the propagation distance) would be divided into mutually exclusive cells, such that a single PB could not be

counted more than once. Cells were stratified into 'positive cells' (staining for viral proteins or dsRNA) or 'bystander cells'. In control treatments (e.g., mock infected or EV) PBs in all cells were quantified. In treatment cells only 'positive cells' were quantified except in Figs 6E (unthresholded) and S9 Fig (no N protein stain). Background staining was reduced using the "Enhance Speckles" function. Due to increased background created by the IF-FISH staining procedure, cells with high background staining in the PB channel were excluded from quantification for S9 Fig. Only DDX6 or Hedls-positive puncta with a defined size (3–13 pixels) and intensity range were quantified using "global thresholding with robust background adjustments" function. All thresholds were consistent within each replicate that used identical staining parameters. PBs were quantified per cell, either control (EV or mock) or treatment (infected or viral protein expressing). For each experiment, an equal number of control and treatment cells were analyzed. For Fig 5, puncta counts were exported and RStudio was used for data analysis and PB numbers in treatment cells were represented relative to control cells for ease of data interpretation, rather than raw PBs per cell. For Fig 9, nuclei and cells were defined according to HUVEC boundaries as specified above. Background of probe specific staining was reduced using a global thresholding strategy with a minimum cross entropy method of execution. The MeasureImageIntensity function was then used to identify the total intensity per image which was divided by the number of cells identified within that image for a final readout of total intensity/cell. A minimum of 40 cells were quantified per condition.

## Statistics

All statistical analyses were performed using GraphPad Prism 9.0. 'Per cell' processing body counts were plotted such that each independent biological replicate (including paired control and treatment) could be visualized on a single graph. Given that per cell processing body counts are naturally skewed and thus non-parametric, we elected to use rank-sum statistical methods (Mann-Whitney U test and Kruskal Wallis test) when appropriate, as indicated in the corresponding figure caption. In the case of Fig 7, a two-way ANOVA with a Tukey's post-hoc analysis was used to determine significance to (1) avoid the false discovery rate associated with multiple $t$-tests and (2) because sodium arsenite-induced processing bodies appeared to be parametrically distributed. Parametric distribution was assumed on all normalized data (Figs 5, 6, 9, 10, S4 and S7) and therefore we elected to use unpaired $t$-tests or one-way ANOVAs with Dunnett's post-hoc analysis, as indicated in the corresponding figure caption.

## Supporting information

**S1 Fig. HUVEC-ACE2 cells are permissive to SARS-CoV-2. A.** HUVECs were transduced with recombinant lentiviruses expressing human ACE2 (HUVEC^ACE2), selected, and infected with SARS-CoV-2 (TO-1 isolate; MOI = 3). 6 and 24 hours post-infection, virus-containing supernatant was harvested and titrated by $TCID_{50}$ assay in VeroE6 cells. These data represent two independent experiments ($n = 2$); mean ± SD. **B-C.** HUVECs were transduced with ACE2 as in A. Cells were fixed and immunostained for DDX6 (B) or Hedls (C) and quantified per field of view using CellProfiler. These data represent two independent experiments ($n = 2$) with >100 cells measured per condition. Each EV and ACE2 replicate pair plotted independently; mean. Statistics were performed using a Mann-Whitney rank-sum test (ns, nonsignificant).
(TIFF)

**S2 Fig. HUVECs are permissive to OC43 and 229E infection. A-B.** HUVECs were infected with OC43 ($TCID_{50} = 2 \times 10^4$) or 229E ($TCID_{50} = 2.4 \times 10^3$). 6-, 12-, or 24 hours post infection,

virus-containing supernatant was harvested and titrated using $TCID_{50}$ assay in VeroE6 (A) or MRC5 (B) cells, respectively. These data represent two independent experiments ($n = 2$); mean ± SD. **C.** HUVECs were infected with 229E as in B or mock infected. Total RNA was harvested 12 hours post infection and RT-qPCR was performed using primers specific to 229E genomic RNA (gRNA) or subgenomic RNA8 (sgRNA8), or HPRT (cellular house-keeping gene). Cycle-threshold (Ct) values for each primer pair were plotted. These data represent three independent experiments ($n = 3$); mean ± SD. Ct values greater than 35 were not deemed biologically relevant and therefore are scored as N/A. **D.** PCR products from C were resolved using agarose gel electrophoresis. gRNA fragment migrated at the expected size of 285 bp. sgRNA fragment migrated at its expected ~80 bp. Representative images from one of two independent experiments are shown.
(TIFF)

**S3 Fig. Hedls puncta are absent in OC43 and 229E infected cells. A.** HUVECs were infected with OC43 ($TCID_{50} = 2 \times 10^4$) or mock-infected. 12 or 24 hours post infection cells were fixed and immunostained for DDX6 (green; Alexa488) and Hedls (white; Alexa555). Nuclei were stained with Hoechst (blue). Representative images from one of three independent experiments shown. **B.** Hedls puncta in OC43-infected or mock-infected cells were quantified per field of view using CellProfiler. **C.** HUVECs were infected with 229E ($TCID_{50} = 2.4 \times 10^3$) or mock-infected. 6 or 12 hours post infection cells were fixed and immunostained for DDX6 (green) and Hedls (white). Nuclei were stained with Hoechst (blue). Representative images from one of three independent experiments shown. **D.** Hedls puncta in 229E-infected or mock-infected cells were quantified as in B. These data represent three independent experiments ($n = 3$) with ≥100 cells measured per condition per replicate. Each mock and infected replicate pair plotted independently; mean. Statistics were performed using a Mann-Whitney rank-sum test (****, $p < 0.0001$). Scale bar = 20 μm.
(TIFF)

**S4 Fig. Processing body protein densitometry after human coronavirus infection. A.** HUVECs were transduced with human ACE2 (HUVEC$^{ACE2}$), selected, and infected with SARS-CoV-2 TO-1 isolate (MOI = 3). Cells were lysed at 6 and 12 hours post infection and immunoblotting was performed using XRN1, Hedls, DCP1A, DDX6, SARS-CoV-2 N, and β-actin specific antibodies. The results of the second independent experiment of two is shown. **B-C.** HUVECs were infected with OC43 ($TCID_{50} = 2 \times 10^4$) or 229E ($TCID_{50} = 2.4 \times 10^3$) or mock-infected. Cells were lysed at 12 and 24 hours post infection (B, OC43) or 6 and 12 hours post infection (C, 229E). Immunoblotting was performed using DDX6, Hedls, XRN1, DCP1A, and beta actin specific antibodies. Protein densitometry was determined in ImageJ normalized to beta actin and expressed as a fold-change relative to the 12 hours post infection (B; OC43) or 6 hours post infection (C; 229E) mock-infected control. These data represent three independent biological replicates ($n = 3$). A one-way ANOVA with a Dunnett's post-hoc analysis was performed; mean; bars represent SD (*, $p < 0.05$).
(TIFF)

**S5 Fig. Ectopic expression of SARS-CoV-2 ORFs.** HeLa cells expressing GFP-Dcp1a were transfected with an empty vector (EV) control or 2X Strep-tagged SARS-CoV-2 ORFs for 48 hours then fixed and immunostained for Strep-tag II (viral ORF; green; Alexa647) or DDX6 (PBs; white; Alexa555). Nuclei were stained with Hoechst (blue). **A.** SARS-CoV-2 ORFs that could be detected by Strep-tag II staining (thresholded ORFs). **B.** SARS-CoV-2 ORFs that could not be detected by Strep-tag II staining (unthresholded ORFs). Representative images from one of three or more independent experiments shown. Scale is the same for all images.

M = membrane protein, S = spike protein.
(TIFF)

**S6 Fig. Validation of processing body loss induced by selected SARS-CoV-2 ORFs in HUVECs. A.** HUVECs were transduced with recombinant lentiviruses expressing 2X Strep-tagged SARS-CoV-2 nsp1, nsp14, ORF3b, and ORF7b constructs or control lentiviruses (EV) and selected with puromycin for 48 hours. Cells were fixed and immunostained for Strep-tag II (ORFs; green; Alexa488) and DDX6 (PBs; white; Alexa555). Nuclei were stained with Hoechst (blue). Representative images from one of three independent experiments shown. Scale bar = 20 μm. **B.** DDX6 puncta were quantified per field of view using CellProfiler. These data represent three independent experiments ($n = 3$) with >18 cells measured per condition per replicate. Each EV and ORF-expressing replicate pair plotted independently; mean. Statistics were performed using Kruskal-Wallis $H$ test with Dunn's correction (**, $p < 0.0021$; ****, $p < 0.0001$; ns, nonsignificant). **C.** HUVECs were transduced as in A. Cells were lysed and immunoblotting was performed using the Strep-Tag II antibody on a 4–15% gradient gel. (TIFF)

**S7 Fig. Processing body protein densitometry after CoV N protein overexpression. A-C.** HUVECs were transduced with recombinant lentiviruses ectopically expressing N protein from the Betacoronaviruses MERS-CoV and OC43 (A), N protein from Alphacoronaviruses 229E and NL63 (B) or SARS-CoV-1 N (C). A control lentiviral expressing an empty vector (EV) was used as a negative control and SARS-CoV-2 N protein expressing lentiviruses were used as a positive control in each experiment. Cells were selected and lysed and immunoblotting was performed using XRN1, Hedls, DCP1A, DDX6, N protein or FLAG, and beta actin specific antibodies. Protein densitometry was determined in ImageJ normalized to beta actin and expressed as a fold-change relative to the EV control. These data represent three independent biological replicates ($n = 3$). A one-way ANOVA with a Dunnett's post-hoc analysis was performed; mean; bars represent SD (*, $p < 0.05$). (TIFF)

**S8 Fig. TNF treatment does not alter processing body number.** HUVECs were treated with 0.01 ng/L soluble TNF for either 6 or 12 hours following which, cells were fixed and immunostained for Hedls. Nuclei were stained with Hoechst. Hedls puncta were quantified per field of view using CellProfiler. These data represent two independent experiments ($n = 2$) with >90 cells measured per condition per replicate. Each mock and infected replicate pair plotted independently; mean. (TIFF)

**S9 Fig. Processing body loss correlates with elevated TNF RNA after ectopic expression of SARS-CoV-2 N protein. A-B.** Cells were treated and immunostained as in Fig 9B and Hedls puncta (white; Alexa488) were quantified per field of view using CellProfiler. Scale bar = 20 μm. These data represent three independent experiments ($n = 3$) with >75 cells measured per condition per replicate; mean. Statistics were performed using a Mann-Whitney rank-sum test (****, $p < 0.0001$). **C-D.** HUVECs were transduced with recombinant lentiviruses ectopically expressing SARS-CoV-2 N or an EV control. Cells were selected, fixed and immunostained for either Hedls (green; Alexa488) and SARS-CoV-2 N (red; Alexa555) prior to hybridization with Stellaris probes specific for GAPDH and TNF (grey; Quasar 670). Nuclei were stained with DAPI. Cells were imaged using a Zeiss LSM 880 laser scanning confocal microscope and the 63X objective. Representative images from two independent experiments are shown. Each channel is shown separately in black and white, including an inverted image

of the RNA FISH channel, prior to merged images. Scale bar = 5 μm.
(TIFF)

## Acknowledgments

We sincerely thank the members of the Corcoran lab, Dr. Craig McCormick and Dr. Eric Pringle for helpful discussions about this work. We would like to thank Dr. Nevan Krogan (UCSF) for the SARS-CoV-2 gene library, Dr. David Proud (University of Calgary) for MRC-5 cells, and Dr. Lorne Tyrrell, Dr. Michael Joyce and Holly Bandi (University of Alberta) for isolates of SARS-CoV-2 variants Alpha, Beta, Gamma and Delta. We would like to thank Dr. Anne Vaahtokari of the Charbonneau Microscopy Facility for microscopy support and Dr. Devender Kumar for CL3 facility support.

## Author Contributions

**Conceptualization:** Jennifer A. Corcoran.

**Data curation:** Mariel Kleer, Rory P. Mulloy, Jennifer A. Corcoran.

**Formal analysis:** Elizabeth L. Castle.

**Funding acquisition:** Jennifer A. Corcoran.

**Investigation:** Mariel Kleer, Rory P. Mulloy, Carolyn-Ann Robinson, Danyel Evseev.

**Methodology:** Mariel Kleer, Rory P. Mulloy, Carolyn-Ann Robinson, Danyel Evseev, Maxwell P. Bui-Marinos.

**Project administration:** Jennifer A. Corcoran.

**Resources:** Arinjay Banerjee, Samira Mubareka, Karen Mossman, Jennifer A. Corcoran.

**Supervision:** Jennifer A. Corcoran.

**Validation:** Jennifer A. Corcoran.

**Writing – original draft:** Mariel Kleer, Rory P. Mulloy, Jennifer A. Corcoran.

**Writing – review & editing:** Mariel Kleer, Rory P. Mulloy, Carolyn-Ann Robinson, Danyel Evseev, Maxwell P. Bui-Marinos, Jennifer A. Corcoran.

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
