## [Decision Letter · Decision Letter 0]

14 Dec 2021

Dear Dr. corcoran,

Thank you very much for submitting your manuscript "Human coronaviruses disassemble processing bodies" for consideration at PLOS Pathogens. As with all papers reviewed by the journal, your manuscript was reviewed by members of the editorial board and by several independent reviewers. In light of the reviews (below this email), we would like to invite the resubmission of a significantly-revised version that takes into account the reviewers' comments.

Dear Authors,

Your manuscript was reviewed by three experts in the field. All three reviewers raise concerns about the set-up, description, interpretation and statistics of some of your experiments. Please have a close look at all of these comments and accordingly adapt the manuscript.

Reviewer 1 accidently had entered his/her minor comments in the " editor-only" box, but after inquiry, he/she has confirmed this was a mistake, and these comments should become available to the authors. I cannot change this in the original review report by reviewer 1, but I will paste the text below for the author's convenience:

(minor comments from reviewer 1:)

1 Figure 1D: authors should add the figure of cells infected with SARS-CoV-2 of 16h post-infection.

2 Figure 1F: authors should show more time of Calu3 cells infected with SARS-CoV-2.

3 Figure 2C and 2E: Most of cells (mock) show no PBs, thus, how can authors prove that OC-43 and 229E truly inhibit PBs.

4 Figure 4: Why do authors only examine the level of mRNA. It is required to detect the level of proteins too.

5 Figure 5D: The expression of nsp4, orf9c, orf10 is not examined.

6 Figure 5: The Immunofluorescent results of other proteins of coronaviruses should be supplied.

7 Figure 6: result show that N protein cannot inhibit PBs induced by Ars. What is the difference between PBs of the normal situation and PBs induced by Ars?

8 Figure 7A and B: Why don't you use SARS's N protein? SARS's N should have a similar phenomenon.(less...)

We cannot make any decision about publication until we have seen the revised manuscript and your response to the reviewers' comments. Your revised manuscript is also likely to be sent to reviewers for further evaluation.

Sincerely,

Marjolein Kikkert

Guest Editor

PLOS Pathogens

Ron Fouchier

Section Editor

PLOS Pathogens

Kasturi Haldar

Editor-in-Chief

PLOS Pathogens

orcid.org/0000-0001-5065-158X

Michael Malim

Editor-in-Chief

PLOS Pathogens

orcid.org/0000-0002-7699-2064

Dear Authors,

Your manuscript was reviewed by three experts in the field. All three reviewers raise concerns about the set-up, description, interpretation and statistics of some of your experiments. Please have a close look at all of these comments and accordingly adapt the manuscript.

Reviewer 1 accidently had entered his/her minor comments in the " editor-only" box, but after inquiry, he/she has confirmed this was a mistake, and these comments should become available to the authors. I cannot change this in the original review report by reviewer 1, but I will paste the text below for the author's convenience:

(minor comments from reviewer 1:)

1 Figure 1D: authors should add the figure of cells infected with SARS-CoV-2 of 16h post-infection.

2 Figure 1F: authors should show more time of Calu3 cells infected with SARS-CoV-2.

3 Figure 2C and 2E: Most of cells (mock) show no PBs, thus, how can authors prove that OC-43 and 229E truly inhibit PBs.

4 Figure 4: Why do authors only examine the level of mRNA. It is required to detect the level of proteins too.

5 Figure 5D: The expression of nsp4, orf9c, orf10 is not examined.

6 Figure 5: The Immunofluorescent results of other proteins of coronaviruses should be supplied.

7 Figure 6: result show that N protein cannot inhibit PBs induced by Ars. What is the difference between PBs of the normal situation and PBs induced by Ars?

8 Figure 7A and B: Why don't you use SARS's N protein? SARS's N should have a similar phenomenon.(less...)

Reviewer's Responses to Questions

**Part I - Summary**

Reviewer #1: PBs are membraneless ribonucleoprotein (RNP) granules found in the cytoplasm of all cells, that mediate decay or translational suppression of cellular mRNAs. PBs or their components are believed to play antiviral roles, however, the impact of which on SARS-CoV-2 infection remains unknow. The authors demonstrate that SARS-CoV-2, OC43 and 229E, induce PB loss. By confocal microscopy and RT-qPCR, authors also found that N protein of SARS-CoV-2 was sufficient to mediate PB disassembly and elevates PBs-regulated cytokine synthesis, thereby reshaping the subsequent immune response. Although these are novel findings and define the impact of SARS-CoV-2 on PBs, but do not truly look at the mechanism by which N inhibits PBs generation. In addition, the direct relation of virus induces PB loss and the elevated transcript levels of selected proinflammatory cytokines needs to be further proved.

Reviewer #2: The manuscript “Human coronaviruses disassemble processing bodies” describes how infection with coronaviruses results in the reduced presence of cytoplasmic puncta that contain known processing body (PB) components in cultured cells. Expression of the N protein from SARS-CoV-2 and not other coronaviruses similarly reduces the presence of PBs in cultured cells. Infection and N expression additionally resulted in the detection of increased transcript levels from a selection of proinflammatory cytokines. The observation that coronaviruses reduce the number of PBs in infected cells is important and the hypothesis that PB disassembly by coronaviruses subsequently increases proinflammatory cytokine transcript levels is very plausible.

I do feel this hypothesis could have been strengthened with the addition of positive controls and have questions regarding inconsistencies in the experimental design and incomplete description of the results and statistical analyses.

Reviewer #3: Robinson et al., have investigated the effects of infection with three human coronaviruses (SARS-CoV-2, OC43 and 229E) on processing bodies (PBs). Previous reports on non-human Coronaviruses (Murine hepatitis virus and transmissible gastroenteritis coronavirus) and PBs have shown virus-specific PB responses. The study by Robinson and colleagues represents an interesting and timely set of descriptive experiments that explore and observe PBs in cells infected with human CoVs. In addition to exploring PB assembly/disassembly in CoV-infected cells some progress has been made to examine viral proteins responsible for PB disassembly and PB-regulated host mRNAs.

While I believe large portions of the work are technically sound, well-executed, and of broad interest, I have major concerns regarding the interpretations of some of the observations and how these observations inform the narrative of PB disassembly. These concerns surrounding the results in figures 4 and 8 reduced my enthusiasm.

**Part II – Major Issues: Key Experiments Required for Acceptance**

Reviewer #1: 1. The direct relation of virus induces PB loss and the elevated transcript levels of selected proinflammatory cytokines is poor. The authors should examine the levels of other mRNAs, such as growth factors and angiogenic factors.

2. The mRNA in the PBs can be displayed in FISH assay, which is a direct evidence authors can prove that the mRNA of inflammation factors is imprisoned in PBs (if it is indeed inside).

3. The cellular factor level of five types of ARE-containing was detected. Why did only three expression levels increase? If the author believes that the increase is because of the depolymerization of PBs, even if it does not detect other types of Are-containing RNA included in the PBs, at least these five should be expressed in similar level (figure 4)

4. Inhibition of PBs might indeed release some mRNA, but I tend to think that these mRNAs are regulatory factors that promote transcription of those cytokines.

5. If PBs play antiviral effect with the infection of the SARS-CoV-2, then will inhibition of the formation of PBs promote viral infection? Authors should clarify this point.

6. The results of immunofluorescence should not only show the picture of merge but also show the separate figure.

Reviewer #2: Major

1. Positive controls. Throughout the introduction you indicate that PB disassembly occurs often during viral infection and in Lines 75-76 you state that: “the presence of PBs correlates with presence/suppression of ARE-mRNAs [7,12-15]”. The addition of a positive control known to disassemble PBs (treatment of viral product) would strengthen the hypothesis that the two observations (loss of PBs and increased levels of cytokine transcripts) are linked. Especially because not all ARE-mRNAs tested display similarly elevated levels upon PB disassembly by coronaviruses or SARS-CoV-2 N specifically. If PB disassembly is the sole reason for the enhanced presence of IL-6, COX-2 and or TNF, should we not expect the same pattern when PBs are disassembled by other treatments?

2. Experimental design is not clearly described. All figures: what is n? are these three independent experiments, three wells, of three ‘field of views’? When fold change is presented (1CE, 2DF, 4, 5BC, 6BDF, 7BD, 8, S1A and S2B) it is not always clear what is depicted. Cells expressing PBs? Or PBs per cell? Figure 5BC describes puncta/cell, so how many cells were analyzed from how many wells in how many independent experiments?

3. Most figures are expressed as fold change (1CE, 2DF, 4, 5BC, 6BDF, 7BD, 8, S1A and S2B). In none of these figures the mock or empty vector that samples are normalized to contains error bars. This suggests that for each data point all samples were normalized to the mock, before the average of n=3 was calculated. If this is the case this removes all variance in the mock group and increases the probability that comparison with a sample population results in significant differences. If the authors insist on presenting fold change, the variance in the mock treatment should be maintained, SDs for mock and EV should be presented and statistics should be performed on the raw data.

4. Immunofluorescent images are used to quantify PBs with a previously established pipeline. The materials and methods describes clearly that quantification with the cellprofiler pipeline was done using consistent thresholds and identical parameters (Lines: 543-544). In some images DDX6 or Hedls containing granules are clearly visible but they do not make the cutoff and are displayed as examples of PB negative cells (e.g. virus infected Zoom images of Figures 1 and 2). To make a fair comparison, not only subsequent quantification analysis but also initial exposure times have to be identical between PB positive and negative (mock and infected) samples. Please provide key information on the exposure times and additionally list the make and model of the microscope.

5. Unclear why in figure 4 COX-2 is examined and found to change significantly upon SARS-CoV-2 infection, while after figure 4 the COX-2 transcript is not mentioned again and for figure 8 TNF is measured. Line 290 even states TNF was elevated after virus infection (Fig.4), while I cannot find TNF anywhere in figure 4. Without clear arguments for this inconsistency, using COX-2 in Fig.4 and TNF in Fig. 8 feels like cherry picking. Furthermore, please clarify why only for Fig. 8 and not Fig. 4 cells were stimulated with TNF.

Reviewer #3: My major concern with the study is the analysis of PB-regulated cytokine mRNAs presented in Figures 4 and 8. The genes tested include IL-6, CXCL8, COX-2, GM-CSF, and IL-1B. These genes are commonly associated with many inflammatory or IFN cascades. Their upregulation and PB-regulation are not definitively linked, and it is more likely that their upregulation is a consequence of IFN/immune signalling than of PB disassociation.

Additionally, in the functional characterisation of these genes in N overexpression (Fig 8), there is no convincing link between the PB disassembly and expression of these genes. Indeed, the expression of these genes is not significantly associated with SARS-CoV-2 N protein overexpression as only one cytokine (TNF) was significantly increased. The title of Figure legend 8 "Ectopic expression of SARS-CoV-2 N elevates selected ARE-mRNAs" is therefore misleading as there is only one upregulated mRNA that is statistically significant. Additionally, the data showing the no effect of TNF treatment on PB numbers should be presented.

It is surprising that authors did not use their Luciferase-ARE reporter assay (included in the pre-print version) to directly link the effect of SARS-CoV-2 infection and protein expression on PB-associated turnover of ARE-containing mRNA transcripts.

For Figure 6: While it looks like there is a difference in PB numbers/fold change between control (EV) and N in Figures 6 D and F, unfortunately, there are no statistics performed. I suppose this may be due to only two replicates of this experiment. At least 3 experiments are required to clarify this. Additionally, it is peculiar that there appears to be slightly more PB assembly in the sodium arsenate treatment when comparing the empty vector and N protein assembly. The reasons for the lack of effect on N on PB disassembly following sodium arsenate treatment need to be at least discussed.

Fig 7. There seem to be lower expression for OC43 and NL63 N proteins than for SARS-CoV-2 which could partially explain lesser effect on PB disassembly. Could authors comment on this? Some form of quantitative analysis of protein expression in western blots in this and other figures (e.g. densitometry of bands) should be presented to justify the conclusions.

**Part III – Minor Issues: Editorial and Data Presentation Modifications**

Reviewer #1: (No Response)

Reviewer #2: Minor

Lines 45-47: Sentence unclear. The relevance only becomes clear after reading the introduction of the manuscript. Moreover, also line 55 describes transcript levels as ‘repressed’. When reading the abstract on its own I am left wondering where? How? and why? Again this all becomes perfectly clear in the introduction of the main text.

Lines 57-60 and lines 108-111: The conclusion that proinflammatory cytokine mRNA levels are increased as an unintended side-effect of SARS-CoV-2 infection is a bit of an odd and counterintuitive note. Firstly, by calling it an unintended side effect it suggests there is no benefit for the virus, leaving no room for tradeoffs with potential positive effects on virus transmission in a natural situation. Secondly, refrain from using the word ‘unintended’ as viruses do not act with intent.

Line 164: Could use a rationale why Fig. 1F is performed at 48 and not 24hpi

Lines 193-194: Fig. 3C, was infection with 229E confirmed?

Figure 5B-D: Viral protein products are labelled differently (E vs env, S vs spike etc)

Figures: abbreviation EV (empty vector) should be explained well before the legend of Figure 8.

Line 548: Check with a statistician whether paired Student’s t-tests are correct. It is my understanding that when comparing cells that have undergone different treatments the test should be unpaired.

Reviewer #3: It is unclear how many experiments were undertaken and what data represents what biological samples for many of the figures.

The convention in the figure legends appears to be set out by the figure legend one for western blots. “Representative images from one of two independent experiments are shown. n=2.” However, only the “n=x” is shown for figures with bar graphs but not pooled for the analysis or mentioned if the trends are the same. If possible, can the results from all experiments be pooled to show the robustness of the data and treatments?

For figure 1, Hedls puncta are quantitated for figure 1 but not for Figure 2. Is there any reason why this was omitted for the other CoVs or can the authors defend the sole use of one puncta quantitation? This is important given that for figure 3, there is quite a pronounced reduction in DDX6 in whole-cell lysates, and while it would make sense there would be less PB machinery, is there the same reduction in Hedls puncta.

Additionally, while it is clear that the DDX6 puncta are drastically less in the Calu cells in figure 1F these have not been quantitated as have the previous figures. Maybe this could be added for the completeness of the figures.

For figure 3 the title is too strong given that one of these viruses does reduce DDX6. For western blots used for comparison and quantitation, could densitometry analyses be performed and statistically assessed.

In figure 5D I am not sure what C145A corresponds to in the western blot, and it is not clear what it is from the text.

Line 444: Origin of FBS.

Line 467: I believe “polyethylimine” should be polyethylenimine

Section Lines 493: Could it be clarified what the target host transcript(s) is/are being used for normalisation.

Line 505: As there are phenotypic differences between isolates and clades of SARS-CoV-2, could the GISAID accession number associated with this isolate be added to the text here? Also, maybe one comment about the lineage designation (Alpha Beta etc).

Line 532: Could the version of CellProfiler be added to this.

Line 594: G should be F

PLOS authors have the option to publish the peer review history of their article (what does this mean?). If published, this will include your full peer review and any attached files.

Reviewer #1: No

Reviewer #2: No

Reviewer #3: No
---

## [Decision Letter · Decision Letter 1]

5 Apr 2022

Dear Dr. Corcoran,

Thank you very much for submitting your manuscript "Human coronaviruses disassemble processing bodies" for consideration at PLOS Pathogens. As with all papers reviewed by the journal, your manuscript was reviewed by members of the editorial board and by several independent reviewers. The reviewers appreciated the attention to an important topic. Based on the reviews, we are likely to accept this manuscript for publication, providing that you modify the manuscript according to the review recommendations.

Your revised manuscript has been reviewed by the three original reviewers. All conclude that the manuscript has improved. However, two of the three reviewers still have concerns, especially about the newly added Fig. 9.

The new figure partly supports the conclusions about cytokine encoding RNAs being present in PBs, however, the conclusion that these are released from PB upon expression of SARS-CoV-2 N protein, as is claimed, is not supported strongly enough by the new data. Reviewer 3 therefore suggests to reconsider the formulation of the conclusions, or add data that does support the claim sufficiently. I agree with reviewer 3 that this is necessary. I also agree with reviewer 2 and 3 that statistical analysis of the new data should be presented more clearly.

Please reply to the reviewers comments to your revised manuscript and adjust the manuscript and figures accordingly.

Sincerely,

Marjolein Kikkert

Guest Editor

PLOS Pathogens

Ron Fouchier

Section Editor

PLOS Pathogens

Kasturi Haldar

Editor-in-Chief

PLOS Pathogens

orcid.org/0000-0001-5065-158X

Michael Malim

Editor-in-Chief

PLOS Pathogens

orcid.org/0000-0002-7699-2064

Your revised manuscript has been reviewed by the three original reviewers. All conclude that the manuscript has improved. However, two of the three reviewers still have concerns, especially about the newly added Fig. 9.

The new figure partly supports the conclusions about cytokine encoding RNAs being present in PBs, however, the conclusion that these are released from PB upon expression of SARS-CoV-2 N protein, as is claimed, is not supported strongly enough by the new data. Reviewer 3 therefore suggests to reconsider the formulation of the conclusions, or add data that does support the claim sufficiently. I agree with reviewer 3 that this is necessary. I also agree with reviewer 2 and 3 that statistical analysis of the new data should be presented more clearly.

Please reply to the reviewers comments to your revised manuscript and adjust the manuscript and figures accordingly.

Reviewer Comments (if any, and for reference):

Reviewer's Responses to Questions

**Part I - Summary**

Reviewer #1: (No Response)

Reviewer #2: This manuscript puts forward an interesting hypothesis on how virus mediated PB loss can increase the availability of cytokine mRNA. The revised version of the manuscript contains considerable improvements in (the description of) experiments and analyses. This clarifies many of my initial questions and certainly improves transparency. With the added RNA-FISH experiment the authors also strengthen their observations. My only comment is that the description and interpretation of this new RNA-FISH experiment can be improved.

Reviewer #3: Most of my queries have been addressed except one of the major concerns (see Major issues)

**Part II – Major Issues: Key Experiments Required for Acceptance**

Reviewer #1: (No Response)

Reviewer #2: (No Response)

Reviewer #3: Authors provide new data (Fig 9B,C) that they claim are showing release of IL6 and TNF mRNAs from disassembled PBs in cells transfected with N-expressing plasmid along with qRT-PCR data of total cell RNA for these mRNAs. While I can accept that Fig 9A does show some co-localization of IL6 and TNF mRNAs in mock cells treated with TNFa, there is no FISH marker for PBs in Fig 9B (as it was in Fig 9A) to justify their conclusion. Yes, there appears to be more TNF mRNA in N-expressing cells but without PB marker it is not possible to conclude that this is the result of TNF mRNA release from disassembled PBs. RT-qPCR of total cell RNA while does show more TNF mRNA (only two repeats though) it doesn't prove that this mRNA is released from disassembled PBs.

I think authors should either provide definitive experimental proofs that cytokine mRNAs are indeed released from disassembled PBs in N-expressing cells or remove this conclusion from the manuscript.

This also refers to their argument against showing data on ARE-containing luciferase reporter assay as RT-qPCR data in the new Fig 9C show no statistically significant difference in IL6 and TNF mRNAs in N-expressing cells.

**Part III – Minor Issues: Editorial and Data Presentation Modifications**

Reviewer #1: (No Response)

Reviewer #2: New figure 9C. it is unclear to me what the errorbars stand for and why there is no statistics. In the discussion you mention that differences in RNA levels (Fig10) are insignificant, though in your rebuttal and main text of the manuscript you do refer to IL6 and TNF as being upregulated and/or having a higher signal intensity. Without statistics I do think you have to carefully formulate this conclusion, especially regarding IL6 as differences both in figs 9 and 10 are small and have considerable variation between replicates. Perhaps doing statistics and reporting the p-values, even when not significant, can make these results better interpretable to readers.

Reviewer #3: Why there is now no PB disassembly in N-expressing cells vs EV control cells in Fig 7D? This contradicts other findings in this manuscript claiming key role of N protein in PB disassembly.

PLOS authors have the option to publish the peer review history of their article (what does this mean?). If published, this will include your full peer review and any attached files.

Reviewer #1: No

Reviewer #2: No

Reviewer #3: No

Figure Files:

Data Requirements:

Reproducibility:

References:

---

## [Decision Letter · Decision Letter 2]

3 Jun 2022

Dear Dr. Corcoran,

Thank you very much for submitting your manuscript "Human coronaviruses disassemble processing bodies" for consideration at PLOS Pathogens. As with all papers reviewed by the journal, your manuscript was reviewed by members of the editorial board and by several independent reviewers. The reviewers appreciated the attention to an important topic. Based on the reviews, we are likely to accept this manuscript for publication, providing that you modify the manuscript according to the review recommendations.

One reviewer still has some concerns about the newly added figure. Please try and address these by adjusting the text and/or adding quantification as suggested.

Sincerely,

Marjolein Kikkert

Guest Editor

PLOS Pathogens

Ron Fouchier

Section Editor

PLOS Pathogens

Kasturi Haldar

Editor-in-Chief

PLOS Pathogens

orcid.org/0000-0001-5065-158X

Michael Malim

Editor-in-Chief

PLOS Pathogens

orcid.org/0000-0002-7699-2064

Dear Dr. Corcoran,

One reviewer still has some concerns about the newly added figure. Please try and address these by adjusting the text and/or adding quantification as suggested.

Reviewer Comments (if any, and for reference):

Reviewer's Responses to Questions

**Part I - Summary**

Reviewer #2: The authors have sufficiently addressed my concerns by adding new experiments in what is now fig 9 and clarifying the methods and statistical analyses. I have no further comments.

Reviewer #3: (No Response)

**Part II – Major Issues: Key Experiments Required for Acceptance**

Reviewer #2: (No Response)

Reviewer #3: While I appreciate that the authors have addressed our concerns with the follow up experiment (Fig 9D,E), to the untrained eye it does not look like there is much of an increase in signal in the bottom panel TNF RNA transcripts in the new Fig 9E (between the two EV and SARS-CoV-2 N bottom panels). I believe we should be seeing more signal or more diffuse RNA transcript signal? Could this be made clearer either through quantitation here in this figure or in a supplementary figure (there is white space under the image that could be used).

With the addition of this experiment and figure it appears that in SARS-CoV-2 N overexpressing cells there is a considerable increase in the Hedls protein level as compared to the EV (Fig 9D,E)? I understand that Hedls protein level was not changed in the SARS-CoV-2 infection at 6 or 12 hours (Fig 4A). Additionally, It doesn’t look like the PBs (appearance of puncta) are reduced or disassembled. Instead it appears that the surrounding signal is stronger suggesting a higher abundance of Hedls rather than disassembly and cytoplasmic redistribution. Is this what was expected from this experiment? Could the authors clarify this observation?

In addition, I raised the concern about lack of demonstrated PB disassembly in Fig 7D, which the authors explained as an outlier experiment. However, new Fig 9D and E also in my opinion don't show PB disassembly (see comment above). Are these data also come from an outlier experiment? This needs to made clear in the text.

**Part III – Minor Issues: Editorial and Data Presentation Modifications**

Reviewer #2: (No Response)

Reviewer #3: A minor issue to address is Dapi should be written DAPI in this figure and others and in multiple places in text.

PLOS authors have the option to publish the peer review history of their article (what does this mean?). If published, this will include your full peer review and any attached files.

Reviewer #2: No

Reviewer #3: No

Figure Files:

Data Requirements:

Reproducibility:

References:

---

## [Editor Report · Decision Letter 3]

4 Jul 2022

Dear Dr. Corcoran,

We are pleased to inform you that your manuscript 'Human coronaviruses disassemble processing bodies' has been provisionally accepted for publication in PLOS Pathogens.

Best regards,

Marjolein Kikkert

Guest Editor

PLOS Pathogens

Ron Fouchier

Section Editor

PLOS Pathogens

Kasturi Haldar

Editor-in-Chief

PLOS Pathogens

orcid.org/0000-0001-5065-158X

Michael Malim

Editor-in-Chief

PLOS Pathogens

orcid.org/0000-0002-7699-2064

The authors have adjusted the text after the second revision to more accurately formulate the conclusions.
---

## [Editor Report · Acceptance letter]

16 Aug 2022

Dear Dr. Corcoran,

We are delighted to inform you that your manuscript, "Human coronaviruses disassemble processing bodies," has been formally accepted for publication in PLOS Pathogens.

Best regards,

Kasturi Haldar

Editor-in-Chief

PLOS Pathogens

orcid.org/0000-0001-5065-158X

Michael Malim

Editor-in-Chief

PLOS Pathogens

orcid.org/0000-0002-7699-2064